# Distinct brain morphometry patterns revealed by deep learning improve prediction of post-stroke aphasia severity
Alex Teghipco [1] ✉, Roger Newman-Norlund[2], Julius Fridriksson[1], Christopher Rorden[2] & Leonardo Bonilha[3]

## Abstract

**Background** Emerging evidence suggests that post-stroke aphasia severity depends on the integrity of the brain beyond the lesion. While measures of lesion anatomy and brain integrity combine synergistically to explain aphasic symptoms, substantial interindividual variability remains unaccounted. One explanatory factor may be the spatial distribution of morphometry beyond the lesion (e.g., atrophy), including not just specific brain areas, but distinct three-dimensional patterns.

**Methods** Here, we test whether deep learning with Convolutional Neural Networks (CNNs) on whole brain morphometry (i.e., segmented tissue volumes) and lesion anatomy better predicts chronic stroke individuals with severe aphasia ($N = 231$) than classical machine learning (Support Vector Machines; SVMs), evaluating whether encoding spatial dependencies identifies uniquely predictive patterns.

**Results** CNNs achieve higher balanced accuracy and F1 scores, even when SVMs are nonlinear or integrate linear or nonlinear dimensionality reduction. Parity only occurs when SVMs access features learned by CNNs. Saliency maps demonstrate that CNNs leverage distributed morphometry patterns, whereas SVMs focus on the area around the lesion. Ensemble clustering of CNN saliencies reveals distinct morphometry patterns unrelated to lesion size, consistent across individuals, and which implicate unique networks associated with different cognitive processes as measured by the wider neuroimaging literature. Individualized predictions depend on both ipsilateral and contralateral features outside the lesion.

**Conclusions** Three-dimensional network distributions of morphometry are directly associated with aphasia severity, underscoring the potential for CNNs to improve outcome prognostication from neuroimaging data, and highlighting the prospective benefits of interrogating spatial dependence at different scales in multivariate feature space.

## Plain language summary

Some stroke survivors experience difficulties understanding and producing language. We performed brain imaging to capture information about brain structure in stroke survivors and used it to predict which survivors have more severe language problems. We found that a type of artificial intelligence (AI) specifically designed to find patterns in spatial data was more accurate at this task than more traditional methods. AI found more complex patterns of brain structure that distinguish stroke survivors with severe language problems by analyzing the brain's spatial properties. Our findings demonstrate that AI tools can provide new information about brain structure and function following stroke. With further developments, these models may be able to help clinicians understand the extent to which language problems can be improved after a stroke.

Aphasia is a language processing disorder that often results from strokes affecting the brain hemisphere dominant for language. It is estimated that aphasia impacts around 30% of all individuals who have survived a stroke[1] and although many individuals experience spontaneous recovery, chronic language impairments are common[2–4], with deficits persisting beyond 6 months in up to 60% of patients[5]. Chronic aphasia is strongly associated with reduced quality of life, more so than stroke alone[6], Alzheimer's, or cancer[7].

Worse aphasic symptoms in the chronic stages have been associated with worse aphasia in the acute period[8] as well as larger lesion volumes and older age at time of stroke[9–11]. Nonetheless, comprehensive models that include lesion characteristics, acute aphasia severity, age and/or demographic information only explain roughly 50% of variance in chronic aphasia severity[10]. This highlights the role of unidentified neurobiological factors in personalized chronic aphasia trajectories.

[1]Department of Communication Sciences and Disorders, Arnold School of Public Health, University of South Carolina, Columbia, SC, USA. [2]Department of Psychology, College of Arts and Sciences, University of South Carolina, Columbia, SC, USA. [3]Department of Neurology, School of Medicine, University of South Carolina, Columbia, SC, USA. ✉e-mail: alex.teghipco@sc.edu

Beyond lesion size, the spatial location of stroke injury is predictive of both aphasia and stroke severity[10,12]. However, modern theoretical models of aphasia neurobiology suggest that language recovery is contingent on the degree of preservation of hierarchically organized neuroanatomical systems beyond the lesion[13–17]. Within this framework, chronic aphasia tends to be less severe when core language specific regions, which support most of recovery, are partially preserved[13]. When language specific areas aren't spared, contralateral homotopic regions, but also right hemisphere regions more broadly, provide the substrate for recovery, typically with reduced likelihood of substantial recovery[13,15,18,19]. More recent work has additionally implicated bilateral domain-general regions in the recovery process[19–21].

Despite growing recognition of the importance of spared regions beyond the lesion, our models for understanding chronic aphasic symptoms have not consistently accounted for the contribution of regional or global brain integrity. Nonetheless, aphasia is commonly associated with atrophy in specific regional networks. For example, Egorova-Brumley et al.[22] reported that individuals with post-stroke aphasia have more atrophy of the inferior frontal gyrus. Stebbins et al.[23] demonstrated that cognitive impairments after stroke were associated with gray matter atrophy in the thalamus, cingulate gyrus and distributed regions across frontal, temporal, parietal, and occipital lobes among stroke survivors with or without mild aphasia. Lukic et al.[24] also demonstrated that preservation of right hemisphere volumes in the temporal gyrus and supplementary motor areas were associated with better language comprehension and production scores among individuals with aphasia. Collectively, these findings suggest that brain tissue integrity beyond the lesion is prevalent and potentially closely linked to the severity of chronic aphasia. However, its importance has only recently begun to receive more serious consideration.

A potential explanation for the relative paucity of studies testing the importance of brain integrity in aphasia are limitations concerning methods that can permit the assessment of the spatial distribution of brain atrophy. Indeed, brain atrophy is known to occur within specific patterns across several neurological conditions, for example, Alzheimer's disease, fronto-temporal dementia, epilepsy, among others[25–27]. In these conditions, specific spatial distributions of atrophy, often affecting the same brain regions, are commonly associated with worse symptoms and outcomes. Such patterns remain largely unmapped in the context of stroke. The emergence of artificial intelligence methods capable of extracting information about *spatially dependent* multivariate features from three-dimensional images affords an unparalleled window into the variance in chronic aphasic symptoms that so far remains unexplained by permitting more robust exploration of heterogenous distributions of brain atrophy in individuals. Beyond atrophy, considering three-dimensional morphometry patterns may make these methods more sensitive to the effects of the stroke on brain regions outside of the immediate lesion (e.g., diaschisis)[28–31], as well as other measures of overall brain health associated with small vessel disease, such as white matter hyperintensities, lacunes and enlarged perivascular spaces[32–35].

Recent work has indicated CNNs can discriminate tissue injury patterns that are not detectable with the same accuracy by conventional machine learning methods[36]. Critically, tissue damage in strokes associated with aphasia typically follows the anatomical distribution of middle cerebral artery (MCA) perfusion territories. While there is some interindividual variability in MCA perfusion territory anatomy, many multimodal brain regions such as those along the midline (e.g., cingulate cortex, supplementary motor areas) or medial temporal regions (e.g., hippocampus, entorhinal and perirhinal cortices) are spared by MCA or anterior circulation strokes. The degree of preservation of these regions is therefore independent from lesion boundaries.

CNNs are increasingly applied to neuroimaging studies, including those in stroke. In the stroke literature, CNNs have been mainly used for lesion segmentation[37–41]. For example, PubMed retrieved 19 studies for the terms, neuroimaging, stroke, and convolutional neural network. Excluding review papers, a study treating Alzheimer's patients and a study that did not use neuroimaging data for modeling, the majority of retrieved studies used CNNs for detecting stroke, segmenting lesions, white matter

hyperintensities, or other regions of the brain that may be affected by atrophy ($N = 12$)[42–53]. Two studies relied on CNNs to accelerate sequence acquisition or increase sequence resolution[40,41], and another two studies leveraged CNNs for outcome prediction, but in the acute setting[54,55]. Nishi and colleagues[56] used a CNN to predict good outcome on the Rankin scale of disability after stroke using Diffusion Weighted Images, finding that the CNN outperformed logistic regression applied to the same data as well as a linear regression that was trained purely on lesion size. Karakis and colleagues[36] report similar findings, showing that a CNN outperformed other classical machine learning methods for predicting type of upper limb motor impairment in stroke using multimodal neuroimaging data. Both studies demonstrate the potential benefits of considering spatial dependence in whole brain neuroimaging data when making predictions about stroke.

In this cross-sectional study, we formulate two hypotheses. First, we propose that a CNN applied to maps of brain lesions and overall brain tissue could outperform standard multivariate machine learning methods for predicting aphasia severity in chronic stroke by directly modeling three-dimensional brain imaging data. Second, we hypothesize that the 3D CNN would take advantage of spatially dependent neuroanatomical information extending beyond the lesion itself. In essence, we anticipate that the CNN will identify subtle patterns of atrophy across the brain—not necessarily limited to canonical language regions—that may be crucial in understanding stroke recovery and aphasia progression. CNNs could thus reveal latent patterns compatible with the quantitative literature that are difficult to detect with traditional methods.

To verify our hypotheses, we develop models comparing CNN with a state-of-the-art classical machine learning algorithm, the Support Vector Machine, which represents the most prevalent machine learning method in stroke neuroimaging and neuroimaging more broadly. We contrast these models' predictive accuracies and investigate the patterns that drive model predictions. Our results show that CNNs achieve significantly higher predictive accuracy than SVMs by effectively leveraging more distributed morphometry patterns. Saliency maps reveal that CNN focuses on both ipsilateral and contralateral brain regions outside the lesion, capturing complex patterns of brain atrophy and integrity that SVM misses. CNN leverages patterns that are consistent among different subgroups of chronic stroke individuals and implicates unique networks associated with different cognitive processes. These findings illustrate the potential of CNNs for advancing our understanding and prediction of aphasia severity following stroke.

## Methods
### Participants

Two-hundred and thirteen individuals (age = 57.98 +/− 11.34, 62% male) with chronic left strokes (mean years post-stroke = 3.2 +/− 3.7) that participated in studies conducted at the Center for the Study of Aphasia Recovery (C-STAR) were analyzed retrospectively in the present work. This cohort represents participants that have been seen at the center through 2022. Data was collected at the University of South Carolina and Medical University of South Carolina. All participants gave informed consent for study participation and the study was approved by the Institutional Review Boards at both institutions (IRB: Pro00053559, Pro00105675, and Pro00005458). Only neuroimaging and behavioral data from participants' first visits was utilized where longitudinal data was collected. All participants had both behavioral and imaging data available for analysis.

### Behavioral assessment

Each participant was administered the Western Aphasia Battery-Revised (WAB-R)[57]. The WAB-R comprises multiple subtests for language impairment in aphasia. The current study utilized the aphasia quotient, which collapses spontaneous speech fluency, auditory comprehension, speech repetition and naming subtest performance into one global score that scales between 0 (reflecting worst aphasia impairment) and 100 (reflecting no aphasia impairment). According to the WAB-R, aphasia severity can be classified into 4 categories using the aphasia quotient: very

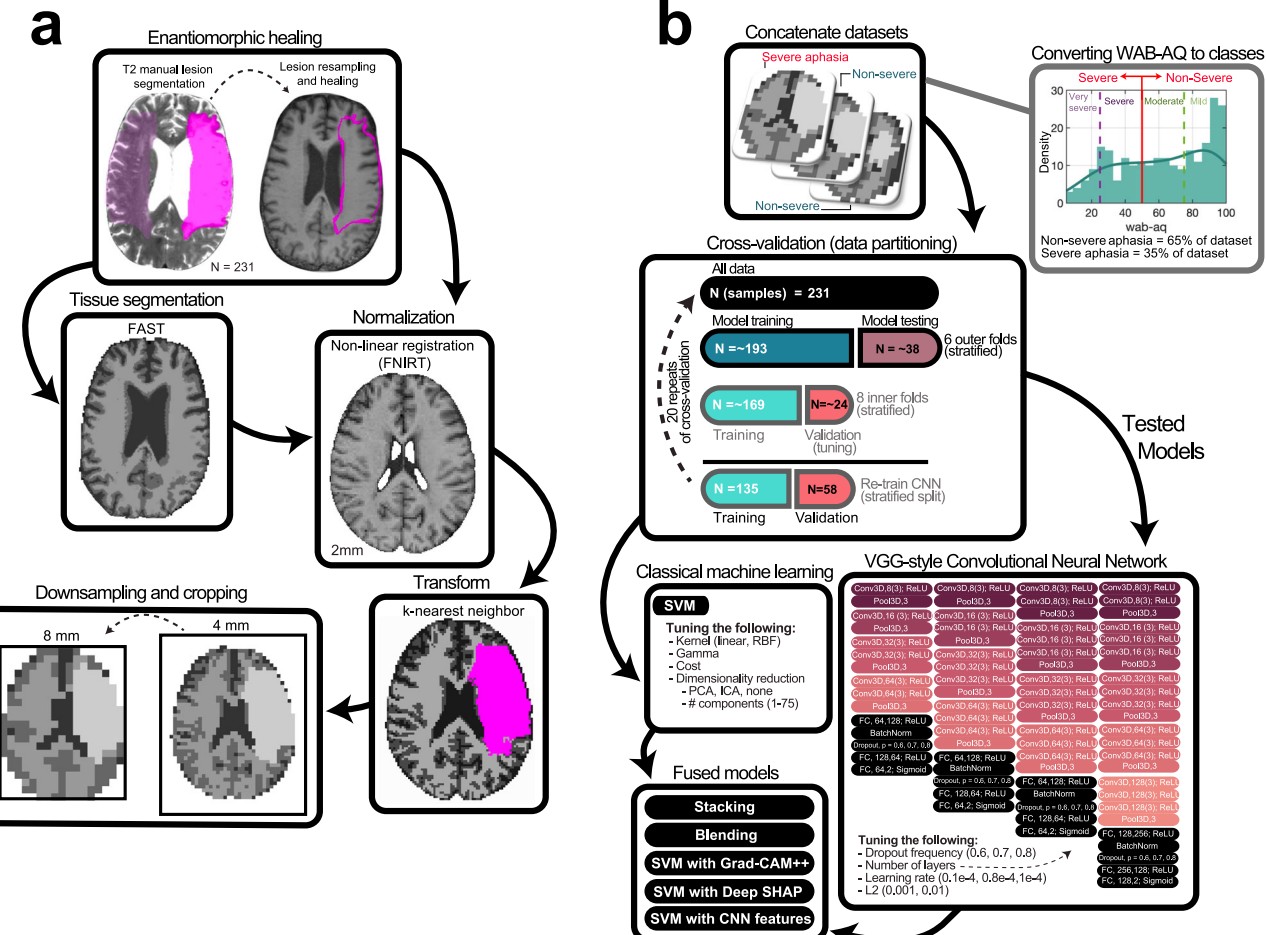

**Fig. 1 | Overview of data preprocessing and analysis. a** Lesion masks (opaque pink; left image) were manually drawn on 231 native T2 scans, resampled to native T1 scans (pink outline right image), refined, and healed by filling them with intact tissue around the homologues (c.f., opaque and transparent pink in image on left; result on right). Middle boxes: Cerebrospinal fluid, white, and gray matter tissues were segmented from healed T1s using FAST (left image). Healed T1s were registered to 2 mm MNI template with FNIRT (right image). Bottom boxes: Tissue and lesion maps were normalized and combined, with lesions superseding other tissue (right image). Volumes were downsampled to 8 mm and cropped (left image). **b** Volumes were concatenated across participants and linked to WAB-AQ, which was used to form severe (35%) and nonsevere (65%) aphasia categories by abridging very severe/severe and moderate/mild categories (denoted by vertical lines on histogram). Data was partitioned for predicting aphasia severity, with model performance evaluated over 20 repeats of a nested cross-validation scheme with stratification (middle box). In each repeat, models were tuned over 8 inner folds, exposing them to approximately 169 samples during training and 24 during testing. Once hyperparameters were selected, the models were fitted to the training data in the outer folds, which consisted of approximately 193 samples. For some models (i.e., CNN), the outer training dataset was repartitioned to leave data for training evaluation. Models were then tested on the approximately 38 samples they had not seen during training or tuning, and the process was repeated for the other outer folds to generate a prediction for each sample in the data. The same partitions were used to train a CNN, SVM, and to implement model fusion strategies (bottom boxes). CNN tuning involved selecting network complexity (see deep learning section in methods for more details), dropout frequency, learning rate and L2-norm (right bottom box; network complexity increases left to right with changes to block composition and/or layer properties). SVM tuning involved selection of kernel, gamma, cost, dimensionality reduction technique to implement prior to training, as well as the number of dimensions to retain. Model fusion entailed averaging predictions made by the two models, stacking the predictions using another model, and chaining CNN-based feature extraction with SVM-based prediction, either by using the learned lower-dimensional features or higher-dimensional saliency maps (SHAP or Grad-CAM++).

severe (0–25), severe (26–50), moderate (51–75), and mild (>76) aphasia. In the present study, we aimed to identify patients with severe aphasia (WAB-AQ below 50), who comprise 35% of the participant cohort and fall under the very severe or severe WAB-R categories. Classifiers were tasked to discriminate participants with severe aphasia from all others (i.e., those with moderate or mild aphasia according to WAB-R). A histogram of WAB-AQ scores across participants is presented in Fig. 1b. Neuroimaging data was collected within 10 days of evaluation on the WAB-R.

### Imaging data
Magnetic Resonance Imaging (MRI) was performed at the University of South Carolina or Medical University of South Carolina using a Siemen's 3T Prisma (upgraded from Trio to FIT in 2016) equipped with a 20-channel RF receiver head/neck coil. T1 and T2-weighted structural scans were utilized in

the current study. A high-resolution T1-weighted MPRAGE sequence was acquired (matrix = 256 × 256 mm, repetition time = 2.25 s, echo time = 4.11 ms, inversion time = 925 ms, flip angle = 9°, 1 × 1 × 1 mm, 192 slices) with parallel imaging (GRAPPA = 2, 80 reference lines). Three-dimensional (3D) T2-weighted sampling perfection with application-optimized contrasts using different flip-angle evolution (SPACE) was used to acquire T2-weighted sequences (matrix = 256 × 256 mm, repletion time = 3200 ms, echo time = 567 ms, flip angle = variable, 1 × 1 × 1 mm, 176 slices) with parallel imaging (GRAPPA = 2, 80 reference lines).

### Image preprocessing
Lesions were segmented manually using T2-weighted images in MRIcron. That is, lesions were drawn by a neurologist (L.B.) or by a researcher with extensive experience with brain imaging in stroke populations. Both were

blinded to behavioral assessments. Lesion masks were resampled to the T1-weighted images using nii_preprocess (https://github.com/rogiedodgie/nii_preprocess/tree/v1.1)[58] and SPM8[59], then refined for any necessary corrections in the case that any additional information about lesion extent was revealed by the T1-weighted image. Anatomical deformation during normalization in the presence of large lesions was avoided using enantio-morphic healing[60]. In this procedure, the lesion boundary is smoothed and the brain tissue inside the smoothed lesion mask is replaced by intact contralateral tissue, thereby exploiting the natural symmetry of the brain to minimize displacement of voxels relative to other methods when normal-izing large unilateral lesions[61]. Healed T1 scans were segmented (binary) into volumes containing white matter, gray matter, and cerebrospinal fluid using FAST[62]. The same healed T1 scans were normalized to the MNI152 (2 mm) template distributed with FMRIB Software Library (FSL)[63] using the fsl_anat pipeline (http://fsl.fmrib.ox.ac.uk/fsl/fslwiki/fsl_anat). Segmented tissue and lesion maps were transformed to template space using k-nearest neighbor interpolation and merged to generate ordinal morphometric maps with voxels inside the lesion being assigned a 4th tissue value. This proce-dure yielded a tissue map with brain structures outside of the lesion and the lesion map itself. The maps were then downsampled to 8 mm voxel size to make our comprehensive cross-validation scheme tractable with deep learning and the field of view was cropped to remove any empty space present across all study participants. Finally, each map was scaled to range between −1 and 1. A visual overview of image preprocessing can be found in Fig. 1a.

## Cross validation

The partitions that we used for training, tuning, and testing deep and classical machine learning models were preallocated to facilitate more equitable paired comparisons (i.e., paired two-sample t-tests) of perfor-mance across 20 repeats of the cross-validation procedure. Repeating cross validation captures the influence of data partitioning noise on the model, which can have substantial impact on performance estimates[64]. Cross validation was stratified to address minority class representation in the partitions. Although we were interested purely in discriminating patients with severe aphasia, the more granular WAB-R aphasia categories were used for stratification, ensuring that partitions sampled data with more similar distributions of raw aphasia quotient *scores* to hedge against the possibility that the model would over or underfit to more specific severity subtypes. Partitions used to tune models were nested to estimate model general-izability more reliably and mitigate the risk of overfitting[65–67]. Six outer folds were used to test model performance, resulting in training sets that con-tained ~192 patients and test sets that contained ~38. Eight inner folds were constructed to tune the models. More inner folds were selected to improve the likelihood of model convergence on an optimal hyperparameter set. Hyperparameters were tested by grid search, using the smallest mean loss over inner folds as the selection criterion. Models were retrained on the entire outer fold using the selected hyperparameters to maximize data exposure. For deep learning, retraining entailed partitioning the outer training set into one training (70%; $N = \sim 135$) and validation (30%; $N \sim = 58$) split, ensuring the test set remains unseen while making validation data available to determine when to cease training.

## Model evaluation

Model performance was captured with precision, F1 scores, and individual as well as balanced class accuracies. These measures were computed by concatenating predictions across outer folds and comparing them to true labels. Individual class accuracies represented the proportion of true posi-tives to false positives with respect to each of the two classes (i.e., severe and nonsevere aphasia). As our task amounts to binary classification, model recall can be reduced to accuracy for the severe class. Accuracy is a highly intuitive measure of model performance but skewed in the presence of class imbalance. For this reason, we computed the mean of individual class accuracies (i.e., balanced accuracy). Model precision captures the fraction of correctly predicted severe aphasia cases (i.e., true positives) out of all severe

aphasia predictions made by the model (i.e., true positives and false posi-tives). The F1 score, which represents the harmonic mean between precision and recall, was prioritized above other measures for model assessment. This score is more appropriate when classes are imbalanced as in our data, and when it is crucial for the model to optimize for both false positives and false negatives. By seeking a balance between these error types, the F1 score ensures that the model does not favor one class over the other. Preference for the F1 score is additionally motivated by the utility of prognosticating patients with severe aphasia, which has the potential to guide more efficient distribution of clinical resources. In this context, false negative predictions are arguably worse than false positives because patients that could benefit from treatment may be missed. At the same time, false positives need consideration as clinical resources are limited. Consequently, we report precision and individual class accuracies to provide better insight into model behavior, but primarily assess models based on the F1 score, and, to a lesser extent, averaged class accuracies.

## Deep learning

Convolutional Neural Networks (CNNs) are designed to learn a hierarchical representation of the data through a series of connected convolutional layers. These layers learn the weights for small filters, or 3-D matrices of weights, that slide over the data through backpropagation, extracting rele-vant features in the process. Blocks of convolutional layers are broken up by pooling layers that reduce the learned feature map dimensionality, increasing computational efficiency of the network and helping to mitigate the risk of overfitting[68]. The network terminates with fully connected layers that transform features into lower dimensional representations and finally into predictions. CNNs are particularly well-suited for neuroimaging data because they do not require flat inputs like other networks and can be extended to use 3D convolutions that are readily applicable to volumetric data. Although some classical machine learning methods have embedded feature selection (e.g., LASSO), CNNs can be more robust in higher-dimensional settings, learn increasingly more sophisticated representations of the data, learn representations that are spatially invariant, and allow the model to account for spatial dependencies (i.e., consider relationships between neighboring voxels at different scales during learning)[68]. This method can improve prediction of aphasia severity through sensitivity to potentially unique patterns of brain morphometry (e.g., atrophy) outside of the lesion (see Fig. 2 for illustration). For a more detailed survey of appli-cations of CNNs to stroke data, see Supplementary Fig. 1 and Supple-mentary Table 1.

We used a single-channel 3D CNN to predict patients with severe aphasia by minimizing binary cross entropy loss weighted by inverse class frequencies to compensate for class imbalance. The network was structured based on VGG (Visual Geometry Group) architecture, using small 3×3 convolutional filters, max pooling, and a deep stack of convolutional layers. This kind of architecture has been shown to perform as well or better than alternatives in similar tasks to ours[69]. During tuning, we systematically experimented with 4 levels of network complexity, allowing the number of convolutional layers and their channel configuration to vary. The least complex CNN structures contained 4 blocks of convolutional layers and the most complex contained 5 blocks. Each block was followed by a max pooling layer with the number of convolutional layers within blocks varying between 1 and 4 and the number of channels in each layer varying between 8 and 128. Networks terminated with three fully connected layers. The first layer doubled the number of channels in the last convolutional layer, the second layer halved the number of channels and the last layer corresponded to the number of classes.

Aside from batch normalization after the first fully connected layer, network regularization was addressed by: i) adding a dropout layer imme-diately after and tuning the dropout frequency (0.6, 0.7, 0.8), and ii) tuning an L2-norm penalty applied to the weight parameters (0.001, 0.01). See Fig. 3b for full network architecture. Tuning was extended to the network learning rate (0.1e−4, 0.8e−4, 1e−4; see supplementary methods for decision on magnitude). To speed convergence and prevent suboptimal

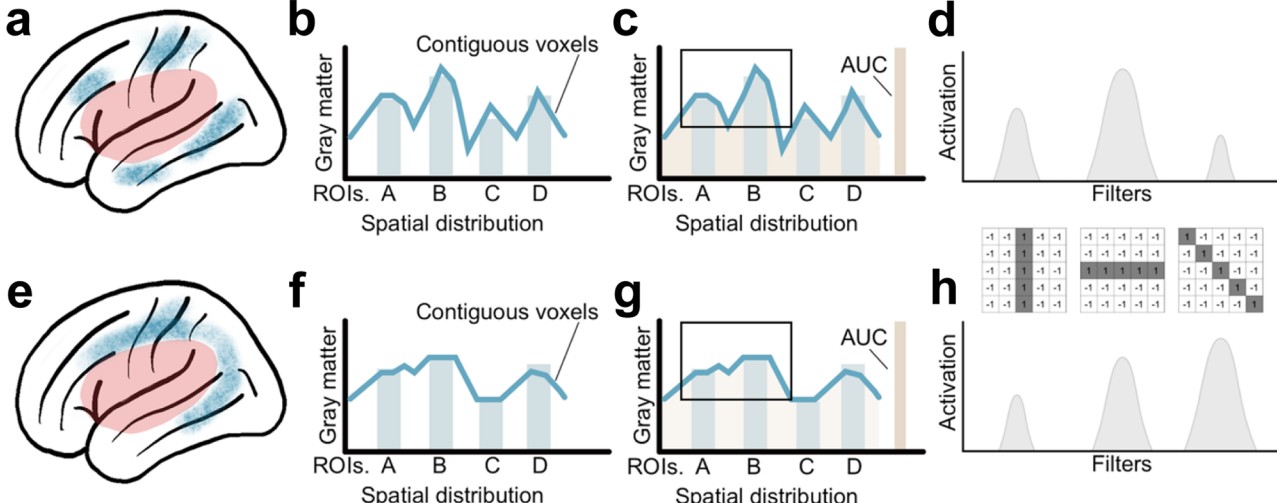

**Fig. 2 | Potential advantages of Convolutional Neural Networks (CNN) for the detection of spatially related changes in brain structure. a** and **e** provide examples of two patterns of gray matter atrophy in perilesional brain areas (shown in blue) of individuals with identical lesions (shown in red). Despite the average voxel-wise atrophy in brain regions being comparable (bars in **b** and **f**), the voxel-wise distribution of the location of atrophy is different, as demonstrated by the lines in **b** and **f** indicating contiguous voxel-wise levels. The total atrophy (area under the curve - AUC in **c** and **g**) is similar as well. While the differences in the pattern of atrophy are visually intuitive (e.g., based on the shape of atrophy in **a** and **c**), they are not captured by conventional statistical models or machine learning approaches. CNNs are ideally equipped to identify such patterns through the application of spatial filters, or weight matrices that slide across the volume to learn features through the process of convolution, and the identification of motifs based on the shape and spatial dependence of imaging features. For example, each filter can represent a spatial contrast, reflecting specialization for different spatial arrangements of voxels, and each contrast can be more or less represented within each pattern of atrophy (**d** and **h**).

solutions a cosine annealing scheduler with warm restarts was used to adjust the learning rate[70]. First, the learning rate was reduced using the cosine annealing schedule, decreasing the rate to 1e−10 over 50 epochs. The process was then restarted. The number of epochs over which cosine annealing was scheduled increased by a factor of 2 with each restart. The CNN was trained and tuned over 800 epochs using mini batches of 128 samples to encourage gradient stabilization. See supplementary methods for information on measures and criteria used to initiate early stopping during training.

**Classical machine learning**

Support Vector Machines (SVMs) were one of the first machine learning methods introduced to neuroimaging[65] and remain the most popular machine learning method in the field (Supplementary Fig. 2). We trained SVMs to predict aphasia severity by minimizing hinge loss weighted by inverse class frequencies using the Sequential Minimal Optimization (SMO) algorithm. In SVMs, the kernel trick efficiently transforms data into a higher dimensional space through which a hyperplane maximizing class separability can be more successfully optimized using different kernel functions[71]. Because optimizing more hyperparameters may lead to model overfitting, we assessed more optimistic estimates of SVM generalizability (i.e., performance) by training independent models using the linear and radial basis kernel functions. Indeed, when we tested SVMs that tuned the kernel function alongside other hyperparameters, performance plummeted (see supplementary results and Supplementary Fig. 3). Other hyperparameters were tuned using random search with 300 logarithmically spaced bins for each parameter. These included kernel scale, which controls the smoothness of the kernel function (ranging from 1e−3 to 1e3), and cost, which controls the width of the margin and balances the trade-off between maximizing the margin and minimizing hinge loss (ranging from 1e−3 to 2e4).

We addressed the possibility that poorer SVM performance relative to CNN reflects optimization failure in high dimensional feature space by cross validating two additional SVMs independently, one which used Principal Component Analysis (PCA) as a data preprocessing step and another that relaxed component orthogonality constraints by applying Independent Component Analysis (ICA) to the components. Reducing the number of features in the data prior to training a model for classification or regression is a commonly employed strategy (e.g., principal component regression) and PCA/ICA are often recommended to be used precisely in this way with SVM in neuroimaging[72]. For both dimensionality reduction techniques, model order selection was expressed as a hyperparameter that was tuned in the inner folds by validating SVM models trained on lower dimensional spaces that retained between 1 and 75 components (i.e., 75 total values tested). Dimensionality reduction was folded into cross-validation—test and validation data were always projected into lower-dimensional spaces defined on training data.

**Fusing classical and deep learning methods**

We considered that SVMs and CNNs may be sensitive to different patterns in the data in several ways. The simplest approach involved computing a weighted average of model prediction probabilities on each test dataset, programmatically adjusting the weight given to one model over the other, and evaluating the resulting ensemble's performance. The next approach integrated model predictions in a more sophisticated way, training a regularized Linear Discriminant Analysis (LDA) to learn the optimal strategy for combining probabilistic model predictions. Tuned CNN and SVM models were used to make out-of-sample predictions in a random half-split of the data reserved for testing in the cross-validation scheme (i.e., each outer fold). These predictions were used for stacking and the remaining test data was used to estimate stacked model performance. Like the weighted average, stacking performance was iteratively evaluated over a range of hyperparameters to provide a better understanding of best-case model performance. Finally, we tested whether prediction performance could be improved by training SVMs to directly exploit the features extracted by CNNs. That is, we cross-validated a SVM that was trained on the CNN's penultimate fully connected layer (i.e., ~64 features in our case).

We further explored the impact of data dimensionality on SVM performance by cross-validating two additional SVM models, one trained on feature saliency maps generated by deep Shapley Additive exPlanations (SHAP) and another trained on feature maps generated by Gradient-weighted Class Activation Mapping++ (Grad-CAM++). This comparison additionally permitted us to evaluate whether it was likely that the CNN

outperformed SVMs by identifying unique patterns in the data unavailable to SVMs. Deep SHAP extends the kernel SHAP framework to deep learning models using an enhanced version of the DeepLIFT algorithm[73,74]. Kernel SHAP leverages Shapley values, which are a game theoretic approach for quantifying the average marginal contribution of a player in a cooperative game[75]. The Shapley value for a feature describes its role in deviating the prediction from the average or baseline prediction with respect to a specific sample in the data. SHAP is an extension of Shapley values that uses the conditional kernel with k nearest neighbors (corresponding to 10% of the samples) for evaluating feature importance[76]. Feature importance is assessed by computing model performance as a function of the inclusion of random subsets of features in the model selected by monte-carlo sampling. The Grad-CAM++ algorithm[77] improves on the original Grad-CAM method of explaining CNN predictions, which involves computing the weighted sum of the gradients of the predicted class score with respect to the feature maps of the last convolutional layer in the network[78]. The Grad-CAM++ algorithm differs by introducing a second-order term into the computation, taking into account the curvature of the CNN decision boundary with respect to changes in feature maps. Some work has indicated that this can improve the reliability and localization of feature importance, including in MRI data, and particularly in the context of complex or nonlinear decision boundaries[77,79]. Grad-CAM++ is also more robust when the CNN contains dropout layers that contribute to more diffuse gradients because it considers spatial correlation between CNN feature maps to derive importance. Fundamentally, Grad-CAM++ is designed to account for the special spatial properties of CNNs that deep SHAP does not directly consider when estimating feature importance. Cross validation of the CNN was implemented in PyTorch[80] and cross validation of the SVM was performed with MATLAB's machine learning toolbox[81].

## Subtyping distinct brain patterns through deep learning

Unsupervised learning of Grad-CAM++ maps was used to explore the heterogeneity of brain patterns learned by the CNN. Feature saliency maps were used as inputs instead of CNN layers to understand which brain structures were implicated. Reliable clustering solutions were identified and generated through the framework of consensus clustering, wherein a resampling technique is used to evaluate clustering consensus[82]. Consensus is computed for each sample pair by measuring the proportion of times the pair is assigned to the same cluster when sampled together. This process results in a symmetric matrix capturing pairwise similarities between samples that reflects whether they can be grouped together consistently. A similarity-based clustering algorithm can be used to retrieve the consensus clustering solution embedded in this matrix[82–85]. The distribution of values within a consensus matrix can be additionally used to understand the internal validity of a solution and guide model-order selection as consensus values provide insight into clustering reliability (see supplementary methods)[82,86].

Here, we subsampled 60% of the voxels in feature saliency maps 1000 times, each time performing k-means clustering using the k-means++ algorithm with 250 replicates to improve solution stability[87]. The eta$^2$ distance measure was used for k-means clustering (i.e., 1 minus the eta$^2$ coefficient). This coefficient aims to capture voxelwise similarity between whole brain maps, with 1 reflecting identical maps, and 0 reflecting maps that share no variance[88]. It shares some resemblance to the Pearson correlation coefficient but quantifies similarity on a point-by-point basis, therefore accounting for scaling and offset when comparing two whole brain maps[88]. For each subsample, clustering was repeated to generate solutions ranging from 3 to 30 clusters. The most complex solution (i.e., highest number of clusters) that exhibited high reliability was selected. Briefly, Hartigan's dip test of unimodality[89] was used to exclude solutions that showed significantly unimodal consensus distributions (i.e., no consensus because samples were not consistently being placed in either the same or different clusters). The proportion of ambiguously clustered pairs[86] was then used to approximate the reliability of the remaining solutions (see supplementary methods for more information). Clusters for the selected solution

were extracted by applying affinity propagation[90] to the corresponding consensus matrix. Affinity propagation is an exemplar-based clustering approach that has been shown to be less sensitive to initialization parameters than other similar methods[90,91]. Critically, an exemplar-based approach provided a representation of each cluster.

## Decoding morphometry patterns

Subgroup-representative feature saliency maps were decoded to probe for functional associations of morphometry patterns predictive of aphasia. Meta-analyses were generated for 200 topics of an author-topic model of the neuroimaging literature[92]. Note, we used the 7th version of the model retrieved with NiMARE[93]. Meta-analyses for the topics were generated with neurosynth[94]. In this framework, a meta-analysis is performed by separating all studies into two groups: those that were associated with a topic and those that were not. Next, a search was performed for voxels where activity was more consistently reported in the topic-associated set of studies. This involved extracting the activation tables from these two groups of studies, creating contingency tables at each voxel that described whether activity was present and whether the topic was associated, and then performing a chi-square test. Decoding was performed by computing the Pearson correlation coefficient between each topic meta-analysis and the saliency map associated with each subgroup after removal of saliency values within the lesion. Topics with a Bonferroni corrected p-value < 0.0001 and correlation above 0.2 were retained.

## Reporting summary

Further information on research design is available in the Nature Portfolio Reporting Summary linked to this article.

# Results

## Evaluating the quality of CNN model predictions and the consistency of learned features

We first evaluated how well a CNN can predict patients with severe aphasia from morphometry and lesion anatomy. Distributions of CNN model performance over 20 repeats of cross-validation indicated the model achieved moderate-to-high accuracy (Fig. 3a). The median accuracy for severe aphasia was 0.88 (ranging 0.81–0.93) and the median of accuracy averaged across the two classes was 0.77 (ranging 0.74–0.78), demonstrating the model was very accurate at identifying severe aphasia and generally accurate at predicting aphasia severity. The model performed less well at identifying nonsevere aphasia cases, attaining a median class accuracy of 0.67 (ranging 0.61–0.69). Precision reflects the same result, highlighting a slight tendency towards false positives to achieve high prediction accuracy for severe aphasia. The precision score of 0.59 (ranging 0.55–0.6) indicated that when the model predicted severe aphasia, it was correct 59% of the time. The model's median F1 was 0.7 (ranging 0.67–0.72) illustrating that it achieved a good balance between precision and recall. For reference, a model that always guesses the majority class would achieve an F1 of 0 and one that always guesses severe aphasia would achieve 0.52. We reiterate our evaluation favored the F1 score, and to a lesser extent, balanced accuracy (see methods).

The robustness of the CNN classifier was tested more formally. Class labels were randomly permuted 500 times and the model building and testing process was repeated. CNNs trained on permuted data produced a distribution of F1 scores with no overlap with CNNs trained on unpermuted data (Fig. 3b). Performance significantly better than chance (p = 0.002 when including the observed result in the null distribution) indicated the network learned meaningful features that distinguish aphasia severity. As further evidence of model robustness, we showed that learned features exhibited a similar structure across cross-validation repeats. T-distributed stochastic neighbor embedding[95] was used to group participants based on features extracted from all models and revealed distinct clusters for predicted classes (Fig. 3c; see supplemental methods for more information). The influence of lesion size (i.e., small, medium and large based on 3 quantiles) and more granular

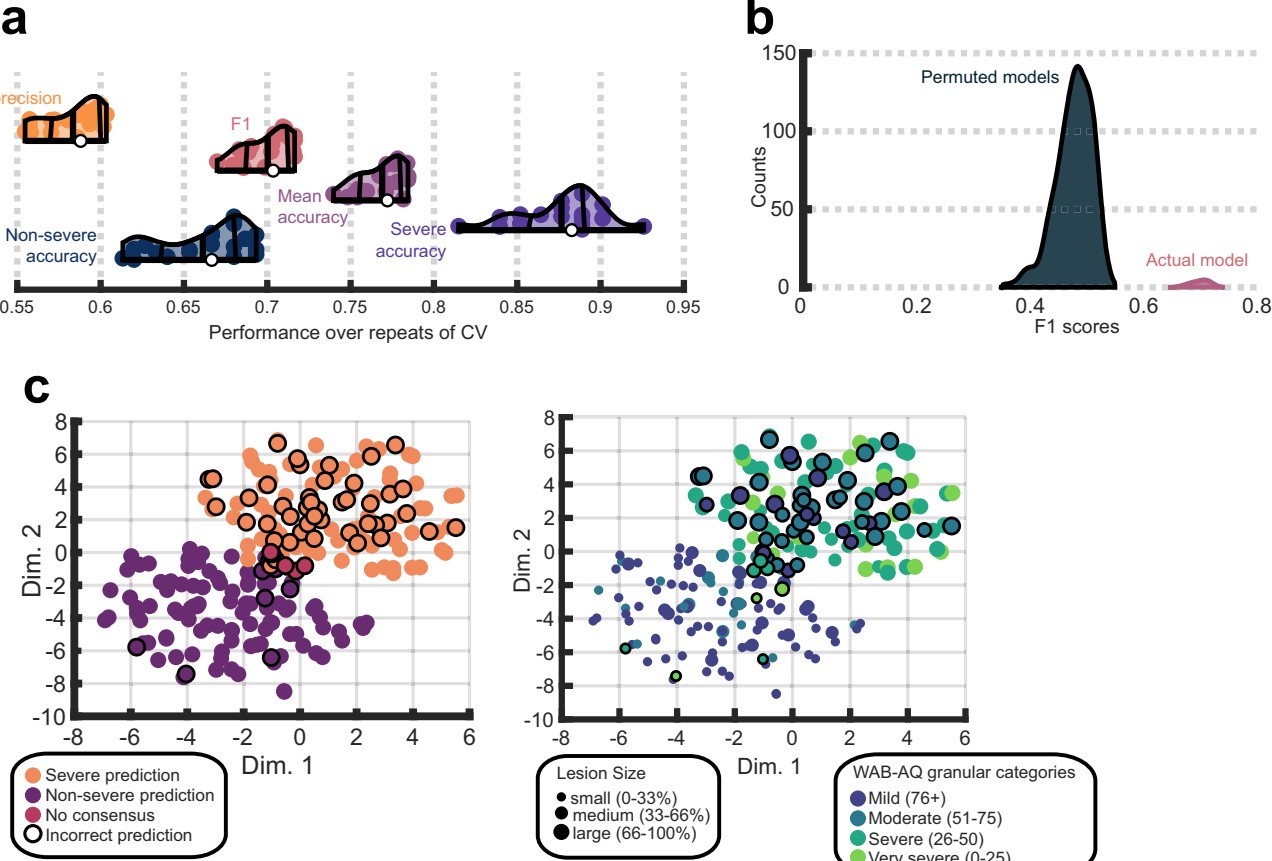

**Fig. 3 | Convolutional neural network (CNN) performance and consistency.**
**a** Violin plots showing CNN performance over 20 repeats of the nested cross-validation scheme. Colored dots represent the result of a single repeat. White dots represent median model performance according to a specific measure. Thick vertical lines inside each violin/density plot represent mean model performance. The flanking thinner, curved, vertical lines represent the interquartile range of performance. Orange violin plots refer to precision, pink to F1 scores, purple to mean or balanced accuracy, violet to severe class accuracy and dark blue to non-severe class accuracy. **b** The entire CNN model building procedure was repeated 500 times, each time permuting the class labels and recording the F1 score for the model during the testing phase. Permuted models' F1 scores are described by the dark blue distribution and the pink distribution shows the unpermuted model F1 scores from **a**. **c** Both

scatterplots show t-distributed stochastic neighborhood embeddings of the first fully connected layers of the CNNs (i.e., concatenating layers across all repeats and folds). Each dot represents a patient. In the left plot, the color of the dot represents the interpolated median prediction made by the CNNs across all repeats of the cross-validation scheme (in 4 patients interpolation did not identify consensus and these patients are designated by pink dots). In the right plot, dots are colored according to more granular WAB-AQ categories that our severe and nonsevere categories collapsed across. Brighter colors correspond to greater aphasia severity. Dot sizes represent relative lesion size using 3 quantile-based lesion size categories (smaller dots correspond to smaller lesions). In both scatterplots, incorrect predictions are distinguished by solid outlines.

aphasia severity categories on the model decision boundary were also investigated (Fig. 3c). Models often predicted nonsevere aphasia when participants had small lesions and severe aphasia when participants had large lesions, reflecting the established relationship between larger lesions and higher severity. Most misclassifications involved participants just below the severe aphasia boundary and medium-to-large lesions.

### Evaluating the quality of CNN model predictions against classical machine learning

We next determined whether SVMs outperformed CNNs (Fig. 4). This could indicate that CNNs, which generally demand more computational resources and involve adjusting more parameters, may have difficulties with overfitting to our sample size and type of data. We first noticed that linear SVMs outperformed nonlinear SVMs across repeats and focus on these models for brevity (see Supplementary Fig. 3). Paired t-tests comparing model performance across repeats revealed that linear SVMs had significantly worse F1 scores (M = 0.65, SD = 0.02) compared to CNNs (M = 0.7, SD = 0.01), t(19) = −10, p = 5.26e−9, Cohen's d = −2.24. Overall, SVMs mean accuracy was worse (M = 0.73, SD = 0.01) than CNNs (M = 0.77, SD = 0.01), t(19) = −9.8, p = 7.28e−9, Cohen's d = −2.2. See

supplementary results for comparisons based on measures we do not prioritize here.

PCA and ICA were inserted into the SVM model building process to address the possibility that SVMs performed worse only as a consequence of the high dimensionality of our data (Fig. 4). Generally, we found that linear SVMs again outperformed other SVMs in this context and focus on the linear models (Supplementary Fig. 3). Paired t-tests revealed usage of PCA (F1:M = 0.65, SD = 0.02; mean accuracy: M = 0.72, SD = 0.02) versus ICA (F1:M = 0.65, SD = 0.02; mean accuracy: M = 0.72, SD = 0.02) conferred no significant performance benefit on F1 scores, t(19) = 0.67, p = 0.51, Cohen's D = 0.18 or mean accuracy, t(19) = 0.61, p = 0.55, Cohen's D = 0.16, suggesting ICA did not capture considerable nonlinearities. Further, dimensionality reduction did not improve SVM performance according to F1, whether using PCA, t(19) = 0.47, p = 0.65, Cohen's D = 0.13, or ICA, t(19) = 0.95, p = 0.35, Cohen's D = 0.3. The same was true for mean accuracy with PCA, t(19) = 0.1.51, p = 0.26, Cohen's D = 0.33, and ICA, t(19) = 1.42, p = 0.17, Cohen's D = 0.45.

We focus on the F1 score but point out that the SVM achieved higher non-severe prediction accuracy while the CNN achieved higher severe prediction accuracy. Despite both models having loss penalized equally by

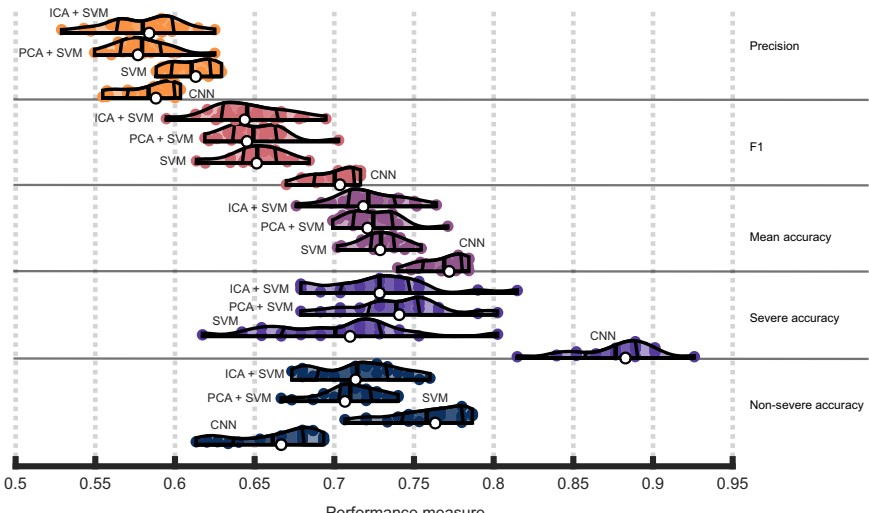

**Fig. 4 | Convolutional Neural Networks (CNNs) outperform classical methods.** Violin plots showing CNN and Support Vector Machine (SVM) performance across 20 repeats of the nested cross-validation scheme. Performance is presented in terms of individual class accuracies (severe and nonsevere accuracy), weighted accuracy (average across the two classes), F1 scores, precision, and recall. Violin plot colors correspond to different performance measures which are additionally separated by horizontal lines. Within each performance measure, the first or topmost violin shows the performance of a SVM combined with an ICA preprocessing step, the following violin plot shows the performance of a SVM combined with a PCA preprocessing step, the penultimate violin plot shows the performance of a SVM without dimensionality reduction as a preprocessing step, and the final violin plot depicts CNN performance as a baseline (i.e., from Fig. 2). See previous figure for information represented in each violin plot.

inverse class frequencies, the SVM tended to have worse accuracy on the minority class, demonstrating a comparatively reduced ability to learn the distinguishing characteristics of this class. While the CNN did tend to make more false positives as reflected in lower precision, balanced measures (F1, balanced accuracy) demonstrated the improvement in recall was more advantageous than the SVMs' better precision.

**Fusing CNN and SVM predictions**

Integrating CNN and SVM models provided us the opportunity to test whether they identified unique patterns. First, we averaged probabilistic class predictions from both models to make final predictions. This analysis demonstrated that exhaustively tweaking the weight given to the CNN over SVM predictions in the average did not produce a single model that meaningfully outperformed the CNN on the F1 score (Fig. 5a). Indeed, a paired t-test revealed no significant difference between the maximum attained F1 from the entire ensemble set (M = 0.71, SD = 0.15) and the CNN (M = 0.7, SD = 0.14), t(19) = 2, p = 0.056, Cohen's d = 0.46. The steep slope of the curve in Fig. 5a highlights the considerable extent to which prediction benefitted from the cumulative accumulation of information learned by the CNN, suggesting that it provides unique information.

The possibility that a more complex criterion for fusing models could improve predictions was investigated by stacking models with regularized Linear Discriminant Analysis (LDA). Limits of stacked model performance were explored by procedurally training and testing the model on each regularization hyperparameter in a large range (500 linearly spaced values between 0 and 1 for gamma; Fig. 5b). On F1 scores, stacked models tended to improve over SVM performance irrespective of regularization choice (c.f., Fig. 5b and Fig. 4). The best-case stacked model blended the SVMs tendency to favor performance on the negative majority class and the CNNs tendency to favor performance on the positive majority class (c.f., Fig. 5c and Fig. 4). While this produced marginally better F1 scores compared to the CNN alone (M = 0.71, SD = 0.03), the difference was not statistically significant, t(19) = 1.2, p = 0.24, Cohen's d = 0.43 (Fig. 5c). The difference was also insignificant when comparing the models' mean class accuracies (M = 0.78, SD = 0.02), t(19) = 1.5, p = 0.13, Cohen's d = 0.59 (Fig. 5c). Just as the comparison between SVM and CNN for Fig. 4, the stacked model improved precision relative to CNN, but had worse recall. When the stacked model was additionally exposed to the inputs of the lower-level models, it

performed worse (Supplementary Fig. 4). Thus, while CNN and SVM models make slightly different patterns of predictions, relatively straightforward ways to stack or fuse models did not identify substantively unique predictive information contributed by the SVM. In contrast, the CNN's comparable performance to the stacked model suggests that it capitalized on more clearly unique predictive patterns.

**Evaluating the impact of CNN properties on feature importance**

Multiple feature saliency mapping methods were used to understand if better CNN performance was the result of the special spatial properties of CNNs. For simplicity, we focused on the mean performing models across repeats of cross-validation but note that learned features exhibited a similar structure across repeats (Fig. 3c). Grad-CAM++ and deep SHAP saliency maps generated for CNNs were compared to SHAP saliency maps generated for SVMs. As only Grad-CAM++ maps are explicitly sensitive to CNN spatial properties, we expected these maps to look different from SHAP and deep SHAP saliency maps, and for SHAP maps to be more similar to each other, provided that the CNN models exploited spatial dependencies. To facilitate model comparisons, we focused on predictions that were accurate for both models.

First, map similarities were compared using the eta$^2$ coefficient (see methods). Similarities were computed between each participant's SHAP and deep SHAP, SHAP and Grad-CAM++, and deep SHAP and Grad-CAM++ maps. Paired t-tests across participants showed SHAP and deep SHAP maps were significantly more similar to each other (M = 0.58, SD = 0.08) than Grad-CAM++ maps were to deep SHAP maps (M = −0.37, SD = 0.16), t(153) = 92.1, p = 6.55e−136, Cohen's d = 4.5, or to SHAP maps (M = −0.47, SD = 0.16), t(153) = 75.9, p = 2.49e−123, Cohen's d = 4. Group-averaged feature maps confirmed these trends (Fig. 6). Grad-CAM++ maps alone highlighted contralateral regions. These were anterior to the concentration of lesions in the group and predicted severe aphasia. Whether predicting severe or non-severe aphasia, SVMs emphasized the region where lesions were concentrated in the group. This was also true of deep SHAP maps generated for CNNs (see Supplementary Fig. 5 for individual maps).

The striking similarity in features that drove model predictions when spatial dependencies were disregarded (i.e., CNN deep SHAP and SVM SHAP), reinforced that it was unlikely for the SVM to contribute

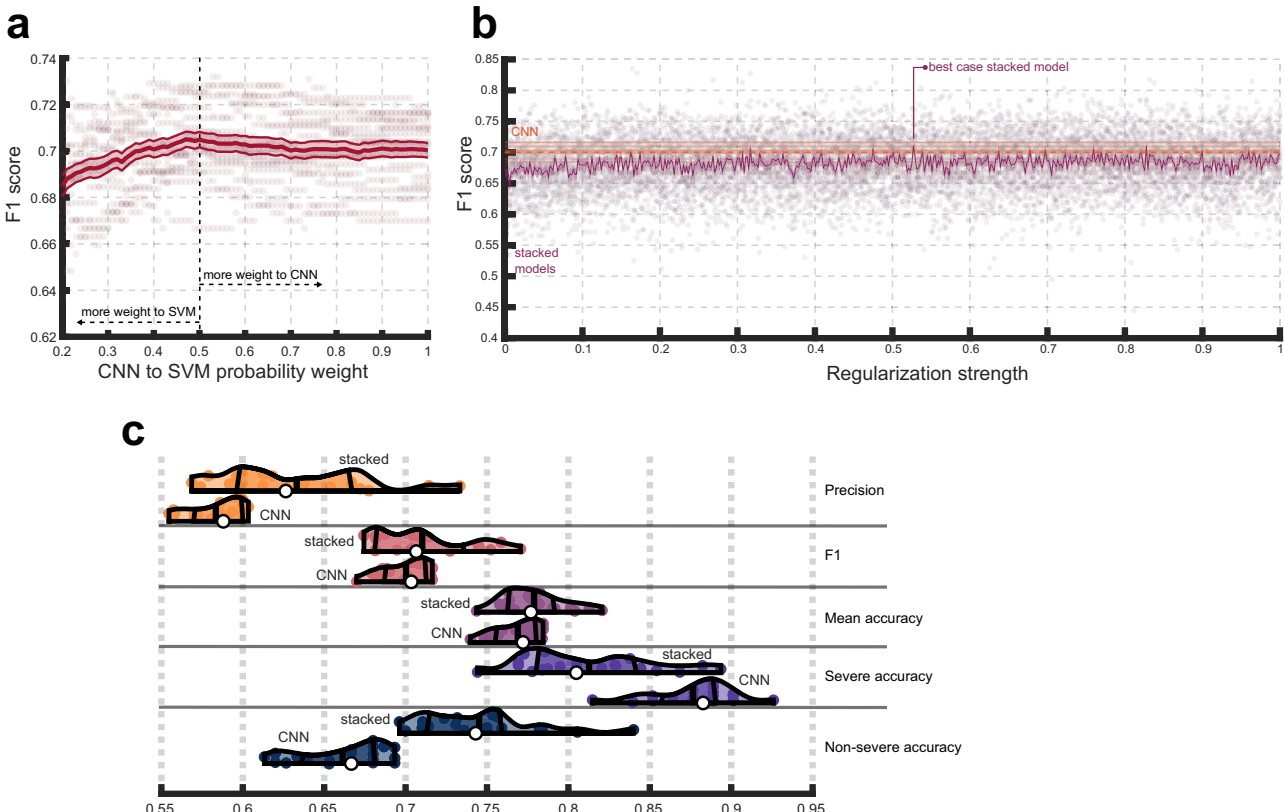

**Fig. 5 | Fusing classical machine learning with CNN. a** F1-scores (y-axis) that result when making final test set predictions by averaging the probabilities assigned to classes by the SVM and CNN models. The weight given to the CNN probabilities over SVM probabilities in the weighted average is depicted on the x-axis (i.e., 1 means only CNN probabilities are considered, 0 means only SVM probabilities are considered, 0.5 amounts to averaging the probabilities without any weighting). The thick solid red line represents the mean F1 score across 20 repeats of cross-validation. The shaded area with thin outer lines corresponds to standard error of the mean across these repeats (SEM). **b** F1-scores (y-axis) are shown as a function of hyper-parameter choice for the stacked model (purple lines and dots) to simulate the best-case stacking result. The solid purple line represents mean performance over 20

repeats of cross-validation with shaded areas corresponding to SEM over these repeats and dots corresponding to results from individual repeats. Performance of the tuned lower-level CNN model is shown for comparison (orange). The thicker dotted line represents mean CNN performance with shaded areas corresponding to SEM and individual thinner and darker solid lines corresponding to results from each repeat. **c** Violin plots showing CNN performance over 20 repeats of the cross-validation scheme (last or bottom violin in each row) relative to the best-case stacked model performance (first or top violin in each row). Violin plot colors correspond to different performance measures which are additionally separated by horizontal lines that separate rows. See Fig. 3 for information represented by the violin plots.

substantially meaningful predictive information to the CNN, echoing previous stacking and model fusion results. CNN emphasis on additional features (i.e., based on Grad-CAM++) supports that it was able to identify unique predictive information relative to SVM, consistent with the finding that fused and stacked models could not significantly outperform it.

If the unique patterns displayed in Grad-CAM++ maps are meaningful and not an artifact or noise, models trained directly on these saliency maps should outperform models trained on SHAP maps. SVMs trained on Grad-CAM++ (M = 0.69, SD = 0.02) achieved significantly higher F1 than SVMs trained on deep SHAP (M = 0.64, SD = 0.03), t(19) = 4.9, p = 9.25e −5, Cohen's d = 1.9 (Fig. 7). Remarkably, training SVMs on these higher-dimensional maps resulted in predictions that were as good as SVMs trained on lower-dimensional CNN features (p > 0.1), which achieved parity with CNN performance (Supplementary Fig. 6).

Having established that qualitative patterns in Fig. 6 are meaningful, we performed a region of interest (ROI) analysis on feature maps from the same models, normalized to have a sum of 1. This analysis examined whether the models' focus on lesioned, perilesional and extralesional regions as well as their homologs varied as a function of predicted aphasia severity (see supplementary methods for additional information). Figure 8 shows that Grad-CAM++ saliency was significantly higher in all left hemisphere regions for nonsevere compared to severe predictions based on two-sample t-tests (all p < 0.0001; see supplementary Table 2 for exact p-values). In contrast,

saliency was significantly higher in all right hemisphere regions for severe predictions (all p < 0.0001; exact p-values can be found in Supplementary Table 2). SVM SHAP saliency maps demonstrated higher feature importance in the lesion for severe prediction, but higher importance in the perilesional and extralesional ROIs for nonsevere predictions (all p < 0.0001; exact p-values can be found in Supplementary Table 2). Feature saliency was low for homolog ROIs across patient class; however, it was relatively higher for severe predictions (all p < 0.0001; exact p-values can be found in Supplementary Table 2). These patterns further establish that CNNs relied on information outside the lesion for successful prediction. This is underscored by a direct test in Supplementary Fig. 7b, demonstrating that training a CNN on lesion anatomy alone leads to significantly worse model performance (p = 0.02).

## Subtyping patterns learned by the CNN

In a final analysis, we mapped the heterogeneity of integrity patterns learned by the CNN, grouping patients according to the similarity of their feature saliency maps by applying consensus clustering to Grad-CAM++ maps generated for all CNN predictions. Separate clustering analyses were carried out for severe and non-severe patient predictions. Subsampling the data revealed strong evidence of cluster structure, with solutions containing 7 and 6 clusters for severe and nonsevere saliency maps displaying high reliability and near unanimous consensus according to the proportion of ambiguously

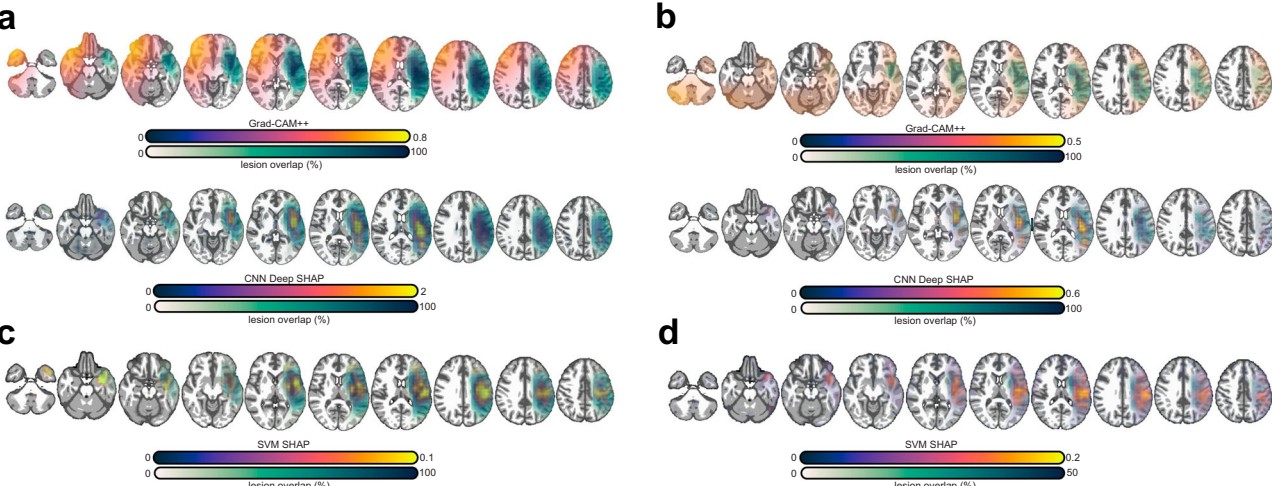

**Fig. 6 | Group-averaged saliency maps. a** Montage in the first row shows mean Grad-CAM++ saliency for patients correctly predicted by the CNN to have severe aphasia (purple to yellow) and their lesion overlap in percentage units (white to dark green) superimposed on a normalized template (neuroradiological convention). Montage in the second row shows mean deep SHAP saliency maps for patients correctly predicted by the CNN to have severe aphasia. Negative SHAP values were replaced with zeros to reflect feature contributions only towards the class predicted by the model. Brighter yellow colors reflect higher feature importance and darker purple colors reflect greater overlap of lesions in the patient cohort. The alpha channels for lesion overlap and mean saliency are modulated by the respective values of those maps to highlight differences between maps. **b** Identical to (**a**) but this montage shows mean Grad-CAM++ saliency for patients correctly predicted by the CNN to have nonsevere aphasia. **c** Identical to previous panels but the montage shows mean SHAP saliency for patients correctly predicted by the SVM to have severe aphasia. **d** Identical to previous panels but the montage shows mean SHAP saliency for patients correctly predicted by the SVM to have nonsevere aphasia.

**Fig. 7 | Grad-CAM++ saliency maps capture unique predictive information.** Violin plots showing that a SVM trained on deep SHAP feature saliency maps (purple) attains poorer F1 scores (x-axis) across 20 repeats of the cross-validation scheme than a SVM trained on the Grad-CAM++ saliency maps (purple), which are capable of capturing spatial dependencies exploited by a CNN.

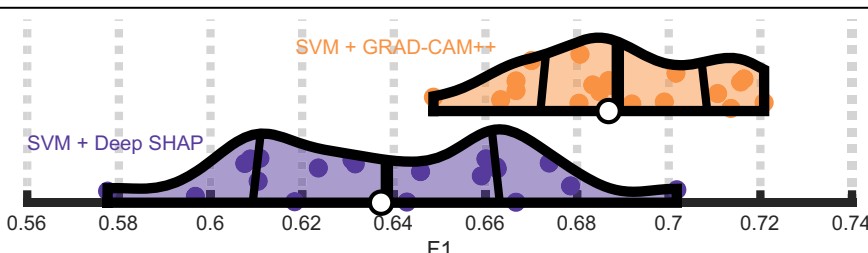

clustered pairs (Supplementary Fig. 8). Analysis of similarity between samples of the same cluster and samples belonging to other clusters reinforced that, overall, clustering distinguished highly unique patterns with individuals displaying high similarity to others of the same subtype (Supplementary Fig. 9). Figures 9 and 10c typify these effects, showing morphology patterns within subgroups exhibited high similarity.

Although group-averaged saliency maps showed strong class differences in lateralization (i.e., left hemisphere for nonsevere and right hemisphere for severe aphasia), subgroups of patients exhibited both patterns, and most subgroups involved more bilateral patterns than suggested by group effects. For example, subgroups 1, 3 and 5 for individuals with severe aphasia showed moderate-to-high saliency in the right hemisphere despite an overall stronger emphasis on the left hemisphere (Fig. 9b). In individuals with nonsevere aphasia, one subgroup showed right-lateralized saliency (subgroup 6) and multiple subgroups showed modest-to-moderate saliency in the non-lateralized hemisphere (subgroups 6, 4, and 2). The right-lateralized subgroup (6) showed remarkable similarity to a subgroup of severe aphasia individuals (2) but was distinguished by more bilateral saliency. Indeed, many subgroups of different classes exhibited overall similar saliency patterns, differing mainly by which hemisphere was more strongly emphasized. As an example, one severe aphasia subgroup (4) displayed highest feature saliency in right temporoccipital cortex, while another nonsevere subgroup (4) displayed peak saliency in left temporoccipital cortex. We note that despite having lower importance, features contralateral to the saliency peak contributed meaningfully to model performance. Ablated SVMs trained on different hemispheres of

saliency maps performed slightly worse than SVMs trained on bilateral maps (Supplementary Fig. 7).

Decoding saliency maps using the wider neuroimaging literature provided evidence that *some* of the variability in aphasia severity is grounded in damage to different language subsystems, but also to a great extent in damage to non-directly language-related systems outside of the stroke, as well as aging (Figs. 9 and 10b, d). First, different portions of the anterior frontal and temporal cortex were associated with different language processes, including semantics (i.e., when anterior temporal pole or posterior inferior temporal gyrus were involved; nonsevere subgroup: 6; severe subgroups: 2,5), reading (i.e., when left posterior temporoccipital cortex was involved; severe:4), spontaneous overt speech (i.e., when left middle frontal gyrus was involved; severe:7), and lexical-semantics (i.e., when all of the aforementioned left hemisphere regions and more of frontal cortex, including inferior frontal gyrus, were involved; severe:9;c.f., Figs. 9 and 10d). Second, decoding showed different portions of superior parietal cortex used for prediction were associated with response inhibition, object and spatial processing, including visuospatial memory and working memory (nonsevere:3,5). Third, patterns targeting anterior frontal cortex around the frontal pole and the temporal pole together were associated with higher-level functions such as decision making, but also age, symptom severity, and a range of brain disorders that included epilepsy and Alzheimer's (severe: 1,3,6; nonsevere: 1,2). For more comprehensive analysis and details about meta-analytic topics, see supplementary results.

Finally, we found that patient subgroups were not associated with lesion size, as hinted by the lesion outlines in Figs. 9 and 10. A one-way

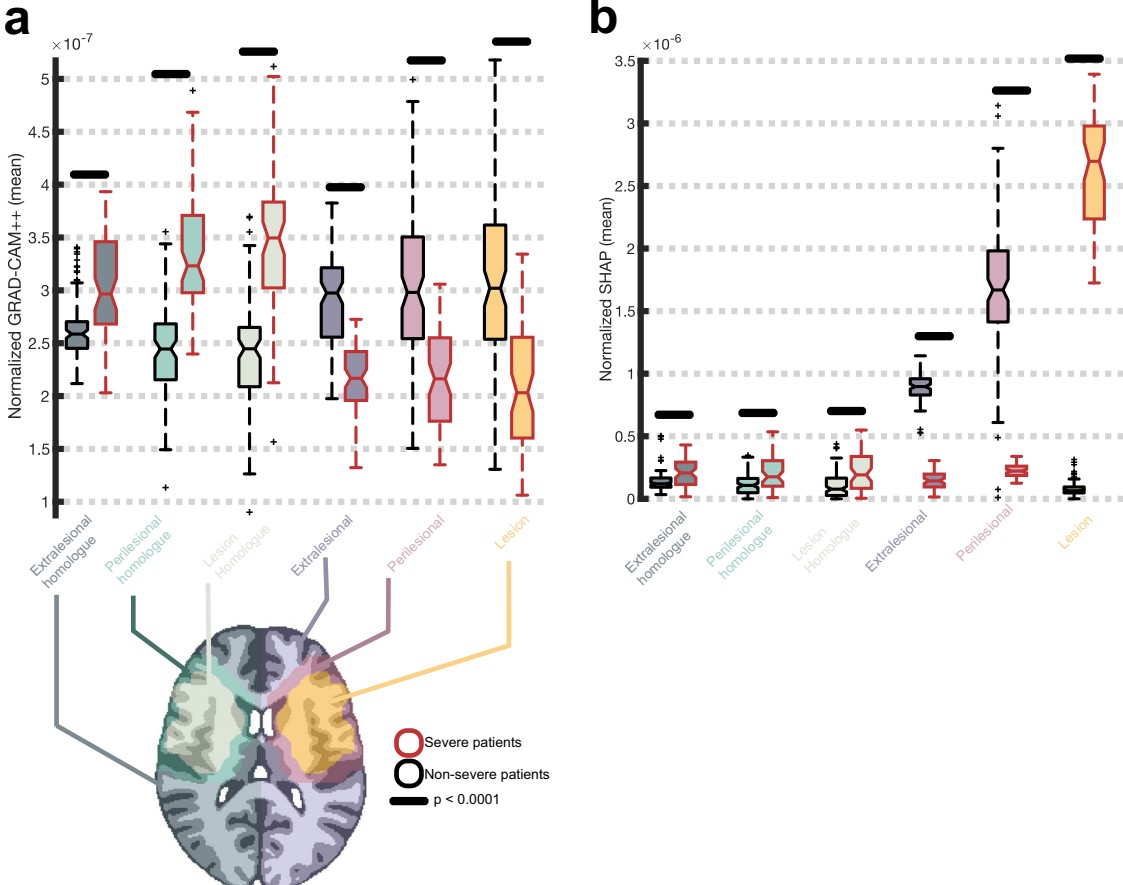

**Fig. 8 | Mean normalized feature saliency within regions of interest. a** Grad-CAM++ feature saliency maps (y-axis) were normalized to sum to 1 and voxelwise values within 6 regions of interest (x-axis; see brain image on bottom of **a** for example visualization in one participant) were plotted as notched box plots independently for patients correctly predicted to have severe aphasia ($N = 69$) and nonsevere aphasia ($N = 104$) by the CNN: the lesion (orange fill), the lesion's right hemisphere homolog (mint fill), the perilesional area (pink fill), the perilesional homolog (light green fill), the extralesional area (i.e., everything in the left hemisphere that's not part of the lesion or perilesional area; dark blue fill), and the extralesional homolog (dark green fill). Each box plot shows the interquartile range (box), median (horizontal solid line), uncertainty around the median (notch width; based on 95% confidence intervals), range (whiskers), and outliers (plus symbols). Severe and nonsevere patients are separated by the color of the box plot lines, with red lines reflecting severe patients and black lines reflecting nonsevere patients. Mean difference between severe and nonsevere patients was tested with two-sample t-tests and horizontal lack lines above box plots indicate significance ($p < 0.0001$). Exact p-values can be found in Supplementary Table 2. **b** Identical to **a** except mean SHAP values (y-axis) are plotted, expressing saliency assigned by the corresponding SVM model for correct predictions of severe ($N = 59$) and nonsevere ($N = 112$) patients. As in the previous figure, negative SHAP values were replaced by zeros before normalization.

ANOVA investigating differences among severe patient subgroups was not statistically significant, $F(6,108) = 0.77$, $p = 0.6$. An identical effect sought for the nonsevere subgroup was insignificant, $F(5,110) = 0.67$, $p = 0.65$. Accuracy was not statistically different among severe, $F(6,108) = 0.53$, $p = 0.78$, and nonsevere subgroups, $F(5,110) = 0.46$, $p = 0.8$ (See Supplementary Tables 3 and 4).

## Discussion

The integrity of tissue outside the lesion offers a rich source of information that stands to improve models of stroke outcome. The current study leveraged deep and classical machine learning to understand how well brain morphometry and lesion anatomy can predict aphasia severity in chronic stroke, testing whether Convolutional Neural Networks' (CNNs) unique sensitivity to spatial properties would identify distinct predictive patterns. In a repeated, nested, cross-validation scheme, we found that CNNs performed well on this task, achieving a median balanced accuracy of 77% and a median F1 score of 0.7. This performance was both significantly better than chance and than the scores obtained by Support Vector Machines (SVMs). Using a variety of techniques for fusing information learned by both models, we found that fused models did not substantially outperform CNNs. A

thorough investigation of feature saliency confirmed that improved CNN predictions were rooted in the identification of more diverse patterns of brain morphology, capitalizing on information outside the area of injury. Clustering the patterns attended by the network showcased diverse aphasia-related morphometry patterns that targeted regions outside the language network and revealed a considerable degree of individualization.

While CNNs can outperform classical machine learning by detecting patterns that other methods are insensitive to, this is not guaranteed—such patterns might be absent and deep learning can be more likely to overfit in small samples[96]. The current study suggests that in modest samples of chronic stroke patients, deep learning can be effective. We used nonparametric testing to demonstrate that CNNs made predictions significantly better than chance, showed that they outperformed classical machine learning methods on our task, and demonstrated that they learned features in a consistent way, exhibiting similar feature structure across hundreds of models trained on different data partitions. We attribute this success to the networks' ability to extract richer latent features from high-dimensional neuroimaging data. Indeed, we found that using more conventional methods of dimensionality reduction, like Principal or Independent Components Analysis (PCA, ICA), did not necessarily improve performance in

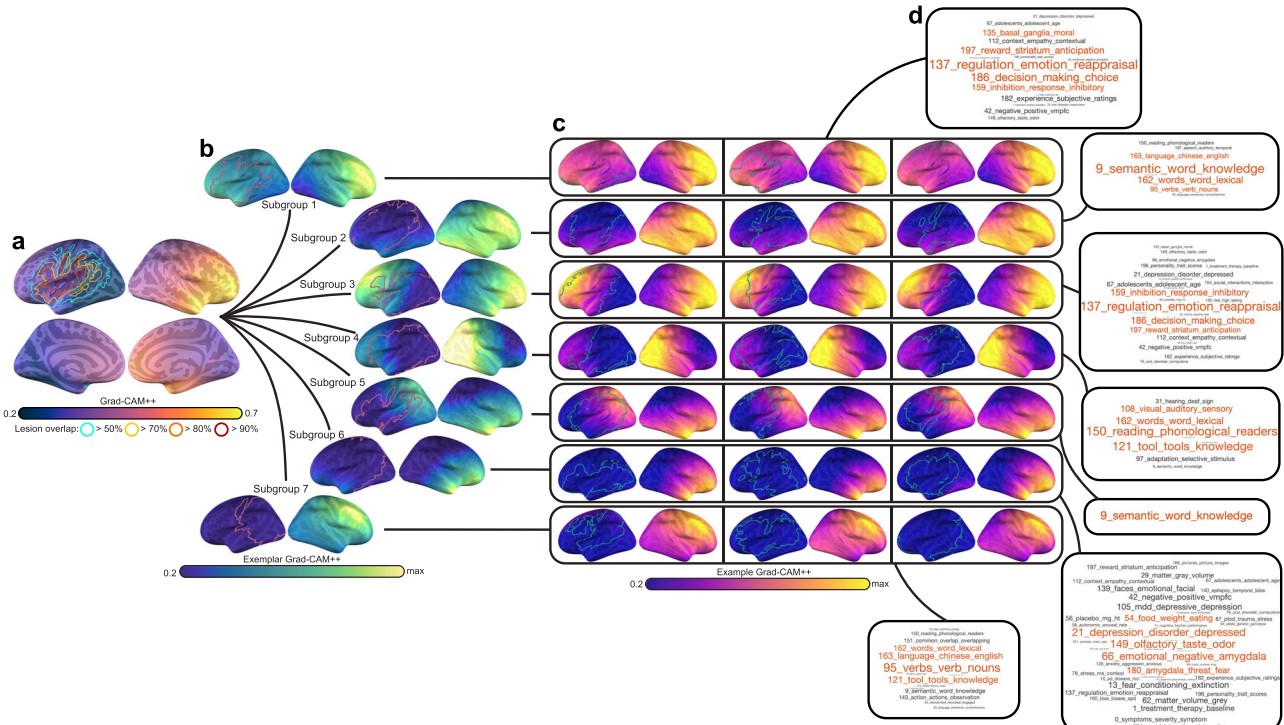

**Fig. 9 | Clustering severe patients using CNN saliency maps and cluster decoding.** **a** Group-averaged Grad-CAM++ feature maps (thermal heatmap) with lesion extent superimposed (outline). Lesion extent is shown for the group based on several percentage thresholds of overlap. **b** Exemplar patients for each patient cluster or subgroup are displayed (viridis colormap) along with each patients' specific lesion map (pink outline). Relative feature importance is shown so the maximum value for each subgroup is different. **c** Saliency maps for three example individual participants that belong to each subgroup are shown (thermal) with their specific lesion maps (green outline), highlighting consistency in feature importance within subgroups. Volume maps were projected onto the fsaverage surface for visualization using RF-ANT[127]. **d** Decoding of subgroup networks (i.e., exemplars) based on Pearson correlation coefficients between extralesional Grad-CAM++ estimates and 200 meta-analyses of topics identified by an author-topic model of the neuroimaging literature. Word clouds show all associated topics with a Pearson correlation above 0.2 (and Bonferroni $p < 0.0001$; exact $p$-values can be found in Supplementary Data 11). Each topic is named based on the 3 individual neuroimaging terms that load most strongly onto the topic. The index of the topic within the model is shown to facilitate cross-referencing the full set of terms[78]. Word size is modulated by the magnitude of the Pearson correlation coefficient. The top 4 associated topics are shown in red.

classical machine learning. However, classical models performed as well as CNNs when trained on lower-dimensional features learned by the networks or higher-dimensional saliency maps capturing the networks' attention. These results corroborate and extend recent work demonstrating that CNNs can be successfully deployed outside their current niche in the stroke literature for lesion segmentation and can effectively predict stroke outcomes better than classical methods[36,56].

One conference paper outside of our PubMed search (see Supplementary Fig. 1 and Supplementary Table 1) has also reported successful application of 2D CNNs but to functional MRI data for predicting language ability in chronic aphasia[97]. Our findings show that CNNs can be applied in a similar capacity to brain integrity patterns. However, our work goes further. In the context of volumetric data, 2D CNNs are less capable of exploiting spatial dependencies compared to the 3D architecture that we deploy here because they ignore an entire spatial dimension (i.e., focusing on individual slices)[98]. We present comprehensive analyses interrogating whether such patterns contributed to prediction of severe aphasia, including direct comparisons between CNN and classical machine learning models trained on the same data and in as similar a way as possible (e.g., definition of partitions, loss weighting, etc). Moreover, our work comprehensively subtypes and attempts to functionally characterize the many different individualized patterns that drive these predictions.

The relative success we were able to obtain with our model, trained on cross-sectional data, hints at the exciting possibility that with continued development and access to larger and more diverse samples, it may be possible to eventually deploy similar models at point-of-care, where prediction of aphasia severity from intake scans may help healthcare professionals prepare patients for their anticipated outcomes and inform

interventions. While an exciting possibility, this potential requires much additional work and will necessitate not just access to larger samples, but also longitudinal data, higher quality scans, and more automated preprocessing solutions among a long list of other essential developments.

The current study used CNNs based on the understanding that they can offer distinct insight into brain morphometry outside the lesion predictive of aphasia severity. The first line of evidence indicating that CNNs exploited spatial properties was that they outperformed SVMs, which did not present clearly unique predictive information during our efforts to fuse the models. Feature saliency maps confirmed this. Group-averaged saliency maps sensitive to CNN spatial properties (Grad-CAM++) highlighted markedly different features than saliency maps that were insensitive to them (SHAP). Compared to SHAP, Grad-CAM++ emphasized a different hemisphere in patients with severe aphasia and showed a high degree of individualization, highlighting regions distal to the lesion in patients with nonsevere aphasia. Critically, SHAP was exceedingly similar between CNNs and SVMs, presenting strong evidence that CNN performance was related to exploitation of spatial information. These patterns were corroborated by quantitative regional analyses, suggesting that often-overlooked spatial dependencies in neuroimaging data may be critical for building more predictive multivariate models.

Post-hoc attribution of attention for CNNs is an active area of research that has seen a proliferation of methods capable of producing slightly different explanations of the model's decision[99,100]. It's critical to note that these methods can be unreliable and there is no single recommendation for all networks[101]. Nevertheless, our results indicated Grad-CAM++ can provide useful explanations. We found that a SVM trained on higher dimensional Grad-CAM++ attributions performed as well as the CNN and significantly

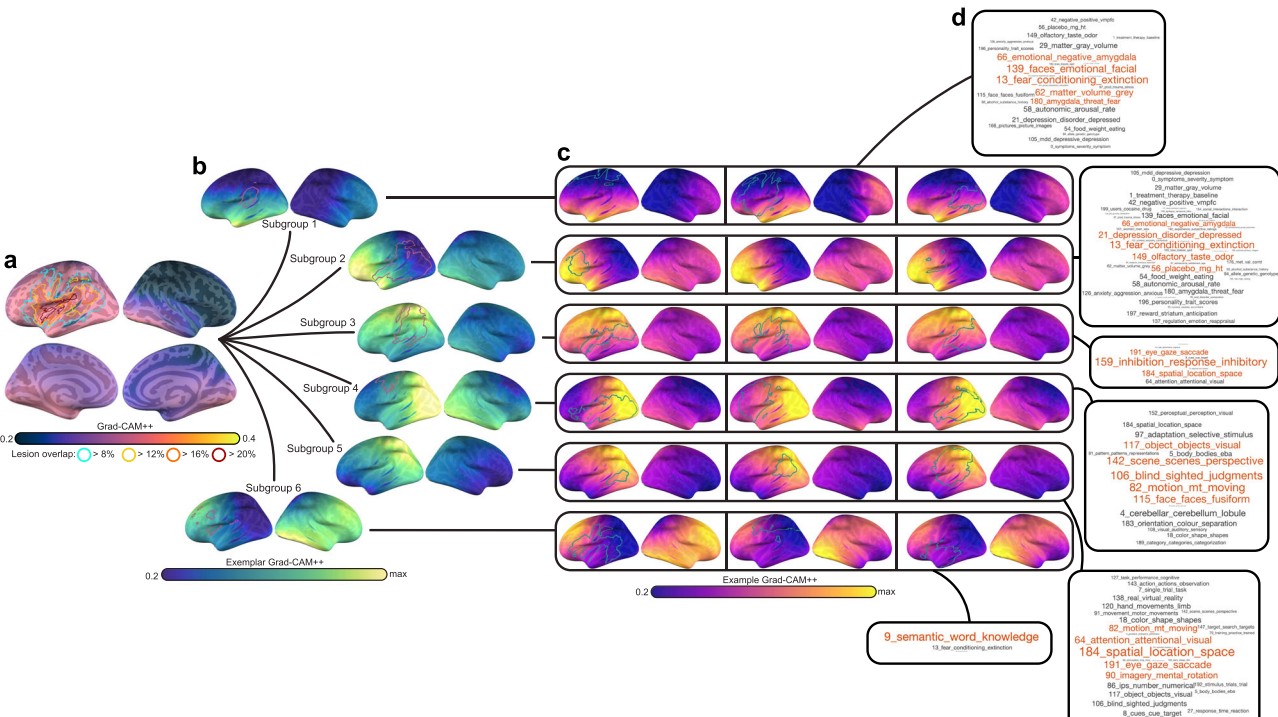

**Fig. 10 | Clustering nonsevere patients using CNN saliency maps and cluster decoding. a** Group-averaged Grad-CAM++ feature maps (thermal heatmap) with lesion extent superimposed (outline). Lesion extent is shown for the group based on several percentage thresholds of overlap. **b** Exemplar patients for each patient cluster or subgroup are displayed (viridis colormap) along with each patients' specific lesion map (pink outline). Relative feature importance is shown so the maximum value for each subgroup is different. **c** Saliency maps for three example individual participants that belong to each subgroup are shown (thermal) with their specific lesion maps (green outline), highlighting consistency in feature importance within subgroups. **d** Decoding of subgroup networks (i.e., exemplars) based on Pearson correlation coefficients between extralesional Grad-CAM++ estimates and 200 meta-analyses of topics identified by an author-topic model of the neuroimaging literature. Word clouds show all associated topics with a Pearson correlation above 0.2 (and Bonferroni $p < 0.0001$; exact $p$-values can be found in Supplementary Data 11). Each topic is named based on the 3 individual neuroimaging terms that load most strongly onto the topic. The index of the topic within the model is shown to facilitate cross-referencing the full set of terms[78]. Word size is modulated by the magnitude of the Pearson correlation coefficient. The top 4 associated topics are shown in red.

better than other SVM models trained on SHAP, raw or dimensionally reduced data (PCA, ICA). The capacity of these saliency maps to predict aphasia severity as well as the latent CNN features provides independent support for their credibility.

Our findings converged on specific features associated with aphasia severity. Identification of individuals with severe aphasia relied on features contralateral to the lesion, whereas identification of nonsevere aphasia relied on ipsilateral features. This global pattern of *relative* saliency illustrates that the CNN identified morphometry in the right hemisphere as a strong predictor of severe aphasia. In the absence of clear morphometry predictors in the right hemisphere, when aphasia was less severe, the network focused on left hemisphere regions. In contrast, SVMs focused on and around the lesion, reflecting to some extent, attention to lesion size. Thus, the discrepancy between CNN and SVM performance signifies the modest but significant gain associated with considering morphology patterns outside of the lesion.

Attention to right hemisphere morphometry is consistent with contemporary models of aphasia recovery, which tend to emphasize worsening aphasia as damage spills out of core language regions and into non-language left and right hemisphere regions[13,18,102]. This view offers a potential reconciliation of conflicting evidence about the contribution of the right hemisphere during language recovery as measured by functional neuroimaging studies[103], by proposing that worse outcomes are the consequence of more extensive left hemisphere damage that necessitates atypical involvement of the intact regions, which happen to be in the right hemisphere. Our findings demonstrate that individuals with severe aphasia additionally have extensive differences in right hemisphere morphometry.

These results contribute to a growing appreciation of extralesional tissue integrity as an independent source of variance in aphasia. It is well-established that coarse information about lesion location and size explains a large amount of variance in acute and chronic stroke outcomes, including aphasia severity[9,11]. At the same time, stroke injury can trigger processes that have a widespread impact on the brain (e.g., neuroinflammation, Wallerian degeneration[104–106]), inducing changes to morphology and functional capacity of regions distal to the injury[28–31]. Recent studies have shown that the accumulation of microvascular injuries, reflected in lower tissue volumes, is linked to both aphasia and response to treatment[32,107]. Further, acceleration of age-related patterns of atrophy independently contributes to aphasia severity[33,108]. These factors may be reflected in the patterns capitalized on by the CNN. Indeed, meta-analysis revealed many saliency patterns were associated with findings from past morphometry studies, including studies on aging. Future work with normative data could more clearly disentangle age-related patterns of atrophy predictive of aphasia severity.

Clustering saliency maps revealed that the relatively localized group saliency observed in individuals with severe aphasia obscured a high degree of individual variability in integrity patterns (7 subgroups). Clustering also revealed more precise morphometry patterns in nonsevere aphasia (6 subgroups), which exhibited diffuse group saliency. Individuals were assigned to subgroups with high reliability, underscoring CNN robustness. Lesion size and model error did not differ among subgroups, further stressing the degree of individualization based on global tissue integrity. That subgroups were more likely to show bilateral effects and sometimes exhibited lateralization patterns contradictory to their group-level trend indicated that any region could be important for predicting aphasia severity within individuals. The heterogeneity of integrity patterns associated with aphasia severity

support a growing body of work characterizing this disorder as a complex of multidimensional deficits[109,110] emerging from varied damage along large-scale networks[111].

Meta-analytic decoding revealed that subgroups targeted different cognitive systems, although more subgroups in severe aphasia demonstrated morphometry patterns associated with activity reported in language studies. Only one nonsevere but four severe aphasia subgroups exhibited saliency associated with language processes. Subgroups emphasized different subsystems, including semantics when anterior temporal pole (TP) or posterior inferior temporal gyrus (pITG) were affected, reading when posterior temporooccipital cortex was affected, spontaneous overt speech when the middle frontal gyrus (MFG) was affected, and lexical-semantics when morphometry differences were widespread across the language network, including aforementioned temporal regions and additional regions in frontal cortex. Curiously, slightly different morphometry patterns affecting largely the same swathe of frontal cortex in combination with the anterior TP were associated with higher-level cognition such as decision making, but also age, prior studies on morphology, and a range of neurodegenerative disorders. These patterns showed no association with language despite being predictive of severe and nonsevere aphasia, suggesting that chronic aphasia may interact with processes reflecting aging and neurodegeneration outside the language system in some patients but not others.

Implicated brain structures were consistent with prior work on the neurobiology of language. For example, subgroups involved frontal cortex, spanning the frontal pole (FP), frontal operculum (FO), inferior frontal gyrus (IFG) and MFG. Prior work in a similar capacity to ours has found that atrophy in IFG discriminates between healthy controls, stroke patients with aphasia and those without[22]. Impairments to different higher-level language functions are observed following damage to IFG[112,113], damage to FO can result in aphasia[114], and resection of MFG can cause apraxia of speech[115]. With respect to regions implicated in temporal cortex, atrophy in TP correlates with semantic dementia[116], stimulation of pITG can disrupt naming[117], and temporooccipital cortex supports reading[118]. Only a minority of saliency patterns were clearly associated with activity in language studies. Thus, while our findings confirm that aphasia severity is linked to morphometry in the canonical language network, they strikingly underline the large extent to which integrity of the brain outside of these regions can contribute. Further, some of the morphometry patterns predicting aphasia severity may be associated with domain-general regions that support language and language recovery after stroke[19–21]. For example, we found that patients with nonsevere aphasia were predicted based on multiple morphometry patterns consistent with functional studies on attention and working memory.

Future efforts at predicting aphasia severity may benefit from improving model precision, aiming to make fewer false positives without compromising accuracy on the minority positive class (i.e., as observed with SVM). Access to larger sample sizes, and especially more patients with severe aphasia, may be helpful for improving performance[119]. We acknowledge that as models improve in accuracy, they may focus on other features than we have highlighted here. We emphasize that our core contribution is that an underexplored source of variance in aphasia can be mapped with deep learning. Moreover, the overall pattern of feature importance we describe converges with other recent sources of evidence that show integrity of areas outside the stroke lesion impact aphasia severity. Ultimately, models such as the one we have presented would have more clinical utility if trained on longitudinal imaging data and we expect these models to converge on similar features.

Our work has focused on segmented tissue maps, but future studies are likely to find better performance through utilization of more granular brain and behavioral data (e.g., using more aphasia classes), as well as the integration of multiple imaging modalities, something that has been recently shown to be effective for stroke outcome prediction with CNNs[36]. Further, integration of demographic and other tabular data is critical for generating the most accurate models possible[120], even though this presents some challenges for training CNNs, which expect spatial inputs. The extent to which inclusion of such data may preferentially boost classical machine learning performance over deep learning is unclear. However, inclusion of non-spatial data will not provide classical methods access to the kinds of spatially dependent patterns we have focused on here.

An important caveat to the results we present is that our data was downsampled to a lower resolution than typical to many neuroimaging analyses to make deep learning with nested cross-validation tractable. Deep learning may not be as effective for higher-dimensional neuroimaging data, other modalities, patient populations, or behavioral measures. Future work aimed to understand the impact of these factors on CNN performance could help inform researchers on the most appropriate tool for their task. Given the impossibility of testing every implementation of classical learning, we also cannot definitively say that these methods cannot compete with CNNs for identification of severe aphasia. Indeed, while our work indicates classical machine learning did not markedly differ in assigning feature importance when the CNN's spatial capabilities were ignored, it is possible that classical learning identified some unique predictive signals that could improve CNN model performance. Our attempts to integrate model predictions assumed that a simple weighted average or linear function could retrieve the optimal model fusion. Further work would benefit from more complex stacking procedures, including setting aside larger data partitions for tuning the stacked model, and stacking more diverse groups of models.

Finally, there is some disconnect between deep learning, which is typically implemented for classification, and prior work on stroke outcomes with classical machine learning, which has focused on regression. While this makes model comparisons with prior work more difficult, here we presented models trained in a generally consistent way with previous work, but framed in the type of classification task that deep learning was designed to excel at. We stress that our findings do not suggest that classical methods (or previously published models) cannot compete with deep learning in terms of prediction accuracy, only that the kinds of patterns that deep learning is uniquely suited to identify are present in neuroimaging data, and useful for prediction, even if they contribute an incremental improvement to model accuracy.

## Data availability

Data from the majority of participants analyzed here are freely available to download as part of our Aphasia Recovery Cohort (ARC) database[121]: https://openneuro.org/datasets/ds004884/versions/1.0.1. A version of the dataset preprocessed in accordance with this manuscript has been made freely available: https://figshare.com/s/46012728175af8029c7b[122]. The minority of individuals used that are currently unavailable in ARC (~20%) will be added to the growing database after it is ensured that the data is processed to comply with HIPAA regulation of Protected Health Information. Nifti brain images used in our figures are available on neurovault (https://identifiers.org/neurovault.collection:16012). Surface renderings from our figures can be downloaded from the above figshare link as a matlab file that can be loaded into our visualization software (see https://github.com/alexteghipco/brainSurfer)[123] to replicate the renderings exactly as we present them (i.e., all colormaps, thresholds, etc. used). Source data is in various files within the Supplementary Information, specific details follow. WAB-AQ scores, shown in Fig. 1b, are available in full within Supplementary Data Set 1. CNN performance measures as depicted in Fig. 3a are available in Supplementary Data Set 2. The raw distribution of permuted F1 scores in Fig. 3b are available in Supplementary Data Set 3. The t-SNE embeddings and patient categories presented in Fig. 3c are available in Supplementary Data Set 4. SVM performance measures depicted in Fig. 4 are available in Supplementary Data Set 5. Performance measures resulting from averaging CNN and SVM model probabilities as shown in Fig. 5a are available in Supplementary Data Set 6. Stacked model performance over regularization parameters, which is plotted in Fig. 5b, is available in Supplementary Data Set 7. The stacked model performances shown in Fig. 5c are available in Supplementary Data Set 8. Performance for SVM models trained on deep SHAP abd Grad-CAM++ saliency maps as shown in Fig. 7

are available in Supplementary Data Set 9. Regional values used to generate all box plots in Fig. 8 are available in Supplementary Data Set 10. Topic-based decodings presented in Figs. 9 and 10 are available in Supplementary Data Set 11. All other data is available from the corresponding author upon reasonable request.

## Code availability

The underlying code for replicating the main models and findings from this study is freely available at: https://github.com/alexteghipco/volDNN[124] and relies on standard functions (e.g., PyTorch). Code for consensus clustering the saliency maps can be accessed here: https://github.com/alexteghipco/consensusClustering[125]. An implementation of the eta$^2$ similarity measure and cluster evaluation measures (proportion of ambiguously clustered pairs and Hartigan's dip statistic) are available as part of this repository. Classical machine learning models were trained using standard functions in the statistics and machine learning toolbox in MATLAB. Code for a pipeline with nearly identical structure to what we employed (including for generating SHAP) can be found in another of our repositories: https://github.com/alexteghipco/StabilitySelection (see live code Tutorial3.mlx)[126]. Any other code is available upon reasonable request to the corresponding author.

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

## Acknowledgements

This work was supported by grants from the National Institute of Health and National Institute on Deafness and Other Communication Disorders (P50 DC014664, U0 1DC011739, R01 DC008355).

## Author contributions

L.B. conceptualized the study with contribution from J.F. and C.R. L.B. designed the study with contribution from A.T. L.B. and R.N.N. supervised lesion segmentation. R.N.N. and C.R. curated data and performed early preprocessing of images. A.T. performed tissue segmentation, registration, and later preprocessing of images. A.T. performed analysis, model development, testing. A.T. and L.B. drafted the manuscript, which was edited and critically reviewed by R.N.N., J.F., and C.R. All authors read and approved the final manuscript and had final responsibility for the decision to submit for publication.

## Competing interests

The authors declare no competing interests.
