## [Peer review File · Communications Medicine]

Reviewer #1 (Remarks to the Author):

This is a paper reporting the design, validation and testing of a variety of post-stroke prognostic models for aphasia. The two key classes of model were: (a) SVMs (representative of 'classical' machine learning); and (b) CNNs (representative of deep learning). The authors find that CNNs out-performed SVMs, when classifying chronic patients as either 'severe' or 'non-severe' aphasic, unless the latter (the SVMs) were trained using features extracted by the former (CNNs). The CNNs also appear to employ features extending beyond the lesion itself in the brain, most notably features representing graded atrophy in the contralateral hemisphere. These extra-lesional features are associated with regions previously associated with language.

This is a good paper, but it is very dense. It is well inside what I consider to be 'my area of expertise', but I still found it quite hard to follow. Most of this is just a natural result of the care the authors have taken to check and re-check the validity and stability of their results - for example by checking that feature importance is broadly consistent across folds of the cross-validation. So I'm not really criticising it, or saying it should be changed. But it's dense.

My outstanding concerns are as follows:

1. The authors say that, as far as they know, this is the first paper to demonstrate CNN effectiveness over-and-above classical machine learning in the area of post-stroke prognoses. I don't want to be 'that guy', but work like this was presented at a 2018 NeurIPs conference: <https://arxiv.org/abs/1811.10520>. I'm not saying the authors should reference it, but I am saying they should soften their claim to novelty (first 'peer reviewed paper describing...? Or something like that).

2. The authors are doing something very interesting as regards using Neurosynth to interpret the extra-lesional regions implicated by the CNN. But while I like it, I'm not really sure it currently does what they say it does. Basically, they say it implicates more language than non-language regions, but to me it just implicates lots of language and non-language regions, and I'd expect the latter result because language implicates most of the brain. If the authors could add some sort of sensible statistical analysis of the Neurosynth associations (language vs not-language), then that section would be stronger.

3. One BIG limitation of this, which is far from lethal but still should be mentioned, is that the authors' models are only using brain data as inputs. Most current post-stroke prognostic models (including some published by these very same authors) also employ non-lesion, tabular data. So by losing this data, they

are presumably lowering the classical machine learning bar that needs to be crossed. This is important because multi-modal deep learning just doesn't work terribly well yet, so this might be what currently prevents CNNs from doing better than classical methods.

4. I also think it's worth mentioning that a lot of post-stroke prognostic work in this field (including lots of excellent stuff that these authors have done) is about predicting scores rather than classifying groups: regression not classification. This just creates a bit of a disconnect between the authors' neat analysis of the literature, to justify using SVMs at all, and the current analysis. It also makes their results hard to compare to the prior art. And again - deep learning just doesn't work very well, in my experience, for regression - another factor that might undermine their core message (that CNNs are better).

5. Related to the same point, I always worry, when binarizing continuous scales, that the authors have basically tried every reasonable cut-off and reported one that works among a sea that don't. Perhaps they didn't try other cut-offs, though I would have in their place, if only to ensure my results weren't artefactual. If they did, and the results are still broadly in keeping with those reported, it would help me if they just said so in a sentence or two. If they didn't, perhaps because other cut-offs are non-sensical in their view, I would appreciate them saying that too.

6. I was not sure about the stacking procedure. Did the authors dispense with all imaging inputs for the stacked model and just use the original models' predictions? If so, then I'm not surprised that the stacking didn't work. It's hard for a model to know the circumstances under which one prediction is more accurate than the other, in this case. The authors' conclusion - that there is no extra info in the SVMs, would only be solid, in my view, if the CNN was Pareto-optimal relative to the SVM: i.e., the CNN got every patient right that the SVM got right, and more besides. If that's true, then one doesn't really need the stacking bit. If it's not true, then I question why the stacking procedure could not find patients where the SVM was right and the CNN was wrong.

7. Finally, I would be a little more cautious than the authors with the claim of near-term practical exploitation. This work depends on time-consuming pre-processing which is semi-automated at best, and which might well be impossible without high resolution, research quality scans. The authors might be confident that stroke clinics can be convinced to take such scans as standard, but I would find that confidence surprising. And if the scans won't be available, we will be stuck with much lower-quality and variable scans, which could well frustrate the kind of analysis used here.

However, even taking all those gripes into account, I think this is a good paper - and particularly good given the promise of open-access data (which I will definitely aim to take up). Can the authors confirm whether they will also share analysis code? There are a lot of steps to this analysis that many of the rest of us would do well to follow, and could follow faster with their help!

Best wishes,

Thomas Hope

Reviewer #2 (Remarks to the Author):

This study compared two methods for aphasia prediction (defined binary as severe and non-severe). CNN on whole brain morphometry (based on segmented tissues) and lesion were compared with SVM (both linear and non-linear). The study population was 231 patients with chronic stroke. CNN achieved higher accuracy and F1 scores than SVM (but not necessarily other model performance metrics). Linear SVM performed better than non-linear, and additional dimensionality reduction didn't improve its performance. Importantly when linear SVM was trained on the latent features learned by the CNN, it performed identical to CNN. Using saliency map, authors suggest CNN leveraged widely distributed patterns of brain atrophy to predictive aphasia severity, whereas the SVM focused on the area around the lesion. However, there were many inter-individual variations. Identification of individuals with severe aphasia relied on features contralateral to the lesion, whereas identification of nonsevere aphasia relied on ipsilateral features.

The method is very well described and appears robust (although this reviewer is not an expert in CNN specifically). The findings are in keeping with the intuition and emerging evidence that whole brain health (in this case segmental atrophy), is associated with post stroke recovery, and extend beyond language specific regions. Using CNN for aphasia prediction is novel and I would like to see it published. Such methods, together with larger studies, are important for developing better aphasia prediction models that help with personalised delivery of care. However, some of the implications may have been over-stated, and there are some errors/clarifications that need to be addressed.

Introduction:

The motivation for focusing on SVM was not clear. Elsewhere in the methods it was stated that "in our experiments, logistic regression and decision trees performed just as well."

When discussing contributions of domain general brain regions to recovery the following key work should be discussed:

- Dynamics of language reorganization after left temporo-parietal and frontal stroke. Brain 2020.

- Domain-general subregions of the medial prefrontal cortex contribute to recovery of language after stroke. Brain 2017.
- Task-induced brain activity in aphasic stroke patients: what is driving recovery? Brain 2014.

Figures:

There are two figures labelled 3. Please correct the labelling and double check all in-text references to all the figures.

Figure 2 is better placed in supplementary material near the Sankey diagram, as its not fundamental to the study.

Wrong figure is cited: A histogram of WAB-AQ scores across participants is presented in Figure 1B.

For Figure 6A, comparing saliency map (Grad-CAM++). Can the authors confirm that the lesion overlap map is not obscuring the view of underlying voxels that have feature importance for this method. Its hard to appreciate any overlay effects. (ie no green/dark blue covering up yellow).

Figure 7: Its helpful to add in the CNN violin plot for comparison.

Figure 8: Left Right orientation of brain slices are opposite to the rest of the figures. Can this be reversed for consistency?

Figure 9: Panels A,B, C are not labelled .

Results:

Even though the authors state they are mainly interested in the F1 score, the 'precision' of the model in predicting severe vs non severe aphasia was barely above chance 0.59. This doesn't seem very effective.

Can the authors comment on why SVM outperform CNN in predicting non-severe aphasia accuracy? Evident in Figures 4 and 5. Relatedly when discussing the Stacking approach , the authors limit the discussion to F1 scores, but do not consider significantly improved performance of CNN in other metrics when stacked with SVM.

Is it valid to compare the saliency maps for the non-severe patients to the meta-analysis data of neuroimaging literature when we see the CNN has a low non-severe aphasia accuracy?

"CNN models often correctly classified participants with medium-sized lesions that belonged to both severity classes", Based on the figure looks like many medium lesions were misclassified. This is misleading.

Discussion:

The prediction of the CNN model is very binary: severe vs non severe aphasia. On a philosophical level, what do the authors see as the actual benefit of a model that can predict severe aphasia from brain imaging when one can do that prediction by simply spending a couple of minutes talking to the patient? What is the impact of such tools? I don't see them impacting treatment decision either, as treatments are impairment based, rather than based on broad diagnostic categorisation.

Therefore I don't think the results shown support this statement : "it may soon be possible to deploy similar models at point-of-care, where prediction of aphasia severity may help healthcare professionals prepare patients for their anticipated outcomes and inform interventions."

Minor:

- Introduction: "interhemispheric territory" > best described as midline brain structures
- The term Neural network is confusing eg : "subtle patterns of atrophy across the neural network". Whole-brain level atrophy would be more clear. Network often is used in context of structural and functional connectivity
- Reference 87 is repeated

Reviewer #3 (Remarks to the Author):

1. Brief summary of the manuscript

This study investigates the potential of deep learning models to predict aphasia severity at the chronic stage after stroke from brain morphometry. The authors show that Convolutional neural networks (CNN) outperform classical machine learning methods such as Support Vector Machine (SVM) in predicting aphasia severity. Information from the brain morphology, especially beyond the lesion location, seems to be a significant contributor to the CNN model performance.

2. Overall impression of the work

This manuscript is a very comprehensive and high-quality investigation. To my knowledge, no similar work of this scale has been published in the field of post-stroke aphasia prediction and I commend the authors for the rigor and extensiveness of the analyses performed and the clarity of the description of

these analyses. This is an important piece of work that provides a compelling precedent for future ML studies on this topic. I have a few suggestions detailed below to enhance the interpretability of this work and facilitate translational applications and I'll mention the four most important here.

One important limitation, which should be highlighted but does not reduce the quality of this paper, is that this work is cross-sectional and therefore does not predict post-stroke aphasia outcomes but aphasia severity at the chronic stage using data obtained at the same stage. This is a common caveat of papers on this topic and the word "predicts"/"prediction" may mislead some readers to think that it takes information at one point in time and predicts information at a future point in time. Although there are not many alternative verbs in ML to describe the results of these models, this limitation should be highlighted in the discussion as it may have significant implications on the capacity of this type of work to be translational. As I mentioned, this work is already a great example to follow for future studies that would explore longitudinal data but this difference needs to be clear because we might expect different findings and interpretations due to changes in brain morphometry (e.g., including the size of the lesion) over time across post-stroke recovery stages if the input of the models are data acquired at the acute stage predicting outcomes at the chronic stage (which is the ultimate goal of this type of work, as stated in "prediction of aphasia severity may help healthcare professionals prepare patients for their anticipated outcomes and inform interventions").

Another point is that the authors highlight that aphasia severity is better predicted by CNN because of the importance of the information beyond the lesion site. If that was the main hypothesis (information beyond the lesion site may improve prediction of aphasia severity), I wonder why the authors did not compare two types of input to the CNN models: one model with only the lesion site information (location and size) and one with lesion information + information from the rest of the brain. While comparing CNN and SVM may lead to this type of comparison indirectly (CNN architecture being more sensitive to whole-brain morphology patterns compared to SVM), this interpretation is indirect and mostly based on post-hoc observations when interpreting the saliency maps. Adding this analysis to the paper would, in my opinion, improve the interpretations of the authors.

Another caveat is that the authors focus all their interpretations and hypotheses around the impact of brain atrophy on aphasia severity. However, as the authors suggest themselves in one paragraph of the discussion, the CNN could also capture other type of information related to 1) the stroke in regions outside of the lesion or 2) (not mentioned by the authors) overall brain health (e.g., lacunes, perivascular spaces). I may have missed something that would relate specifically the CNN models to atrophy but if there is not a clear direct relationship between the two, the authors may need to rephrase their introduction and discussion to not limit their main interpretations to patterns of atrophy and extend it to overall brain health ("morphology patterns outside the lesion" is also a good alternative used by the authors in some places). Related to the previous point, statements like these two seem speculative: "Our findings demonstrate that individuals with severe aphasia additionally have extensive atrophy of the right hemisphere" / "CNN identified atrophy in the right hemisphere as a strong predictor of severe aphasia" (the manuscript only shows that information from the anatomy, in a broad sense, not just atrophy, of the right hemisphere is important to classify individuals with severe aphasia).

Finally, although accuracy was not the main measure to assess model performance, the authors may add more nuance in their interpretations and could mention in the results and discussion the non-trivial higher performance of the stacked model for the non-severe group in terms of accuracy (Figure 5B) as it

seems to indicate that SVM may actually capture some information about the non-severe group that CNN does not.

3. Additional specific comments

- In the introduction, please add citations related to this sentence: "Beyond lesion size, the spatial location of stroke injury is predictive of aphasia severity".

- The paragraph following "Within this framework, chronic aphasia tends to be less severe when core language specific regions, ..." in the introduction does not explain accurately the framework cited. The recruitment of domain-general regions and its role in this theoretical hierarchy has not been described in this framework. This theory only mentions secondary centers in the ipsilateral network and perilesional areas. I agree however that, since this framework, multiple studies have hypothesized a role of domain-general regions in aphasia recovery alongside right-hemisphere regions. I suggest the authors provide more recent citations that support these hypotheses and rephrase this paragraph to match the right hypotheses with the respective papers and avoid claims on a hierarchy between the recruitment of domain-general and right-hemisphere regions (or provide citations for this).

- I may have missed this information but if it is not there, the authors need to mention the range and average of months post-stroke onset for the population included in these analyses.

- The authors may want to briefly explain what filters and contrasts mean in the context of CNN for clarity.

- The numbering of the figures needs to be rearranged (there are two figures 3, and this sentence does not point to the correct figure: "A histogram of WAB-AQ scores across participants is presented in Figure 1B")

- The authors mention in the discussion that downsampling to 8mm voxel size is a limitation but do not explain why they made this choice in the first place as it seems other options were available. It would be good to add this explanation.

- In Figure 3 (the first one), the authors may want to simplify and clarify the box related to VGG-style CNN by reducing the amount of text and highlighting the important information.

Response to reviewers

Reviewer #1 (Remarks to the Author):

This is a paper reporting the design, validation and testing of a variety of post-stroke prognostic models for aphasia. The two key classes of model were: (a) SVMs (representative of 'classical' machine learning); and (b) CNNs (representative of deep learning). The authors find that CNNs out-performed SVMs, when classifying chronic patients as either 'severe' or 'non-severe' aphasic, unless the latter (the SVMs) were trained using features extracted by the former (CNNs). The CNNs also appear to employ features extending beyond the lesion itself in the brain, most notably features representing graded atrophy in the contralateral hemisphere. These extra-lesional features are associated with regions previously associated with language.

Author response: *We thank Dr.Hope for the careful and insightful review, as well as the supportive comments throughout. We believe we have addressed the points raised and remain open to any further adjustments that may be necessary.*

This is a good paper, but it is very dense. It is well inside what I consider to be 'my area of expertise', but I still found it quite hard to follow. Most of this is just a natural result of the care the authors have taken to check and re-check the validity and stability of their results - for example by checking that feature importance is broadly consistent across folds of the cross-validation. So I'm not really criticising it, or saying it should be changed. But it's dense.

Author response: *We appreciate this feedback and have trimmed some text, as well as moved some figures and text into the supplemental materials. Of course, we have also added text to integrate the feedback we have received in this review and remain open to improving the accessibility of our work further.*

My outstanding concerns are as follows:

1. The authors say that, as far as they know, this is the first paper to demonstrate CNN effectiveness over-and-above classical machine learning in the area of post-stroke prognoses. I don't want to be 'that guy', but work like this was presented at a 2018 NeurIPs conference: I'm not saying the authors should reference it, but I am saying they should soften their claim to novelty (first 'peer reviewed paper describing...? Or something like that).

Author response: *Thank you for pointing us to this work—we apologize for having missed that reference and are glad that other studies have appreciated the potential of CNNs in chronic stroke outcome prediction. We have softened the claim of novelty in both the discussion (see below) and introduction (removal of a sentence that had a similar effect). As a small point of clarification, we do not claim that our paper is the first to show CNNs can outperform classical machine learning in post-stroke prognoses, but rather chronic aphasia.*

Further, we now reference the 2018 NeurIPs study (page 22). We also highlight some differences between that study and ours—for example, the use of 2D vs 3D CNNs as well as a different neuroimaging modality for prediction (morphometry vs functional MRI). Finally, we highlight some ways in which our work builds on the prior study, including by subtyping saliency

patterns and attempting to connect them to the broader neuroimaging literature, as well as directly comparing CNN performance to classical machine learning methods in the same data and using as similar of an approach as possible.

2. The authors are doing something very interesting as regards using Neurosynth to interpret the extra-lesional regions implicated by the CNN. But while I like it, I'm not really sure it currently does what they say it does. Basically, they say it implicates more language than non-language regions, but to me it just implicates lots of language and non-language regions, and I'd expect the latter result because language implicates most of the brain. If the authors could add some sort of sensible statistical analysis of the Neurosynth associations (language vs not-language), then that section would be stronger.

Author response: *The reviewer raises an important point, and we agree that the text regarding the interpretation of the saliency maps and the decoding analysis was somewhat confusing. We have now clarified that section.*

Specifically, our interpretation was not that the decoding implicated more language than non-language regions. Indeed, as the reviewer indicated, many saliency patterns were associated with non-language functions. We have clarified this in the results (page 19) as well as the discussion (24-25). This interpretation supports our overall message carried from the introduction that overall brain integrity can contribute to aphasia severity.

We note that our results are presented regionally for convenience as there is overlap among the regions highlighted in the saliency maps, but we have tried to clarify that we are not making regional claims about language specificity.

Nonetheless, salient areas important for predictions of severe aphasia were associated with more language-related topics compared to non-severe aphasia (4/7). Therefore, individuals predicted to have severe aphasia have altered morphometry broadly consistent with language-related activation patterns. The difference in magnitude of association to language-related topics vs non-language-related topics in Figures 9 and 10 provides a visual display of differential associations, which aid in the interpretation of the results.

To further this point, we explore the brain function topics associated with the 4 saliency patterns related to severe aphasia (data depicted in Figures 9 and 10 of the manuscript). The key observation is that for most of these patterns, only topics clearly related to language were significantly associated. In the few cases where other non-language related topics showed some association, there was clear overlap in the cognitive processes reflected by the group of topics (as reflected in the font size in Figure 9). These were:

1. *Semantic_word_knowledge*
 - *This topic is undisputedly related to language and no other topics showed a significant association*
2. *Reading_phonological_readers, Speech_auditory_temporal, language_chinese_english, semantic_word_knowledge, words_word_lexical, verbs_verb_nouns, language_sentences_comprehension*
 - *All of these topics are undisputedly related to language*

3. *Words_word_lexical, Reading_phonological_readers, Hearing_deaf_sign, visual_auditory_sensory, Tool_tools_knowledge, semantic_word_knowledge, adaptation_selective_stimulus*
 - *Note, visual_auditory_sensory and adaptation_selective_stimulus clearly reflects the paradigms being used in many of the studies on this topic*
 - *Tool_tools_knowledge is consistent with semantics and related to language. We note that the full list of terms mapping onto this topic includes semantics*
4. *Verbs_verb_nouns, language_chinese_english, words_word_lexical, reading_phonological_readers, semantic_word_knowledge, action_actions_observation, recruitment_recruited_engaged, language_sentences_comprehension, common_overlap_overlapping, verbal_fluency_overt*
 - *The majority of these topics are language-related. Some topics reflect common but uninformative terms used by the group of studies identified (e.g., recruitment_recruited_engaged). Action_actions_observation shows association as there is considerable interest in the shared neural substrates of action perception and language processing (e.g., the full list of terms that load onto this topic includes embodied). Again, we would emphasize that it is interesting enough that so many clear language-related topics show strong association.*

We appreciate the reviewer's comment about the lack of clarity regarding the Neurosynth comparisons and we have now clarified that our intent was simply to compare the saliency maps with an atlas-based definition of functional localization in the brain from fMRI studies. As the list above suggests, the saliency maps associated with worse aphasia were indeed related to brain networks commonly activated in language functions, thus confirming that spatially dependent patterns noted by CNN can be therefore interpreted as a form of lesion mapping where the language network combined structure is taken into account. Since this is the type of information that CNN (and only CNN) can integrate as a whole, this is the value added by CNN.

3. One BIG limitation of this, which is far from lethal but still should be mentioned, is that the authors' models are only using brain data as inputs. Most current post-stroke prognostic models (including some published by these very same authors) also employ non-lesion, tabular data. So by losing this data, they are presumably lowering the classical machine learning bar that needs to be crossed. This is important because multi-modal deep learning just doesn't work terribly well yet, so this might be what currently prevents CNNs from doing better than classical methods.

Author response: *These are excellent points we have worked to integrate into the manuscript. Our limitation section had previously discussed that models like ours that include multi-modal data will likely fare better. Indeed, we reference a study throughout the introduction and discussion that has shown better prediction of stroke outcomes with multimodal imaging data used as input to a CNN. This study is also referenced in our limitation section and we have added some text to further highlight it (page 25). While our limitation section also mentioned that improving the granularity of the behavioral data would be beneficial, we have additionally clarified that inclusion of demographic and other tabular data is critical for generating more accurate models (page 25). We hope the development of approaches for integrating tabular data with CNN models that expect spatial data will generate models that capture a wider spectrum of predictive signal available. We also caution, however, that the addition of non-*

spatial data will not make classical machine learning methods sensitive to the kinds of spatially dependent patterns we have emphasized in our work here.

4. I also think it's worth mentioning that a lot of post-stroke prognostic work in this field (including lots of excellent stuff that these authors have done) is about predicting scores rather than classifying groups: regression not classification. This just creates a bit of a disconnect between the authors' neat analysis of the literature, to justify using SVMs at all, and the current analysis. It also makes their results hard to compare to the prior art. And again - deep learning just doesn't work very well, in my experience, for regression - another factor that might undermine their core message (that CNNs are better).

Author response: *This is another great point that we now discuss in the limitations section (pages 25-26). As the reviewer points out, CNNs were originally designed for classification. For this reason, we implemented classical machine learning in a generally similar way to previous work while framing the problem in terms of classification. We appreciate that regression-based problems are an ultimate goal of AI in general, and we believe that this study offers an important first step towards that objective.*

6. I was not sure about the stacking procedure. Did the authors dispense with all imaging inputs for the stacked model and just use the original models' predictions? If so, then I'm not surprised that the stacking didn't work. It's hard for a model to know the circumstances under which one prediction is more accurate than the other, in this case. The authors' conclusion - that there is no extra info in the SVMs, would only be solid, in my view, if the CNN was Pareto-optimal relative to the SVM: i.e., the CNN got every patient right that the SVM got right, and more besides. If that's true, then one doesn't really need the stacking bit. If it's not true, then I question why the stacking procedure could not find patients where the SVM was right and the CNN was wrong.

Author response: *We appreciate this overall concern, and we agree that our point was not clear. The main observation is that CNN is sensitive to unique information compared to SVM. We have carefully edited the manuscript to emphasize this point (e.g., pages 14,16 and 25-26).*

We also clarified that SVM saliencies do not readily improve CNN performance in a substantial way (e.g., pages 14,16,22-23 and 25). Our support for this from the main text is the following:

- 1. The SVM does not attend substantially different features to the CNN when we ignore spatially dependent patterns (i.e., SHAP values in Figure 6).*
- 2. A simple and intuitive model fusion approach of taking the weighted average of class prediction probabilities (Figure 5A) shows that, at best, a fused model performs marginally but not significantly better than the CNN.*
- 3. An additional analysis we have added (pages 14-15 and new panel B in Figure 5) shows that model stacking using a wide range of all possible hyperparameters produces at best a solution that does not outperform the CNN (consistent with the weighted average results)*

The point in (3) reflects a newly introduced analysis where we attempted to simulate the best-case stacked model tuning scenario by training and testing the stacked model over the entire range of hyperparameters that are available. This analysis shows that even under more ideal conditions, the stacked model would likely be bound by CNN's performance. Note, the stacked

models we present in the revised manuscript were refined to avoid data leakage during the training procedure. Previously, we had used in-sample predictions for the lower-level models. This change, described on page 9, boosted stacked model performance on the independent test datasets and established performance in line with the results from the model fusion approach (i.e., simple weighted average of prediction probabilities).

Stacking may produce similar or worse performance if the additional model does not contribute new, complementary, or sufficiently distinct information. Moreover, while there is a difference in pattern of errors between the two models, the SVM tends to predict the majority class despite the fact that both SVM and CNN loss was penalized by inverse class frequencies. The persistent focus of the SVM on the majority class indicates that its error pattern might reflect a strategic bias aimed at enhancing prediction accuracy, which masks the model's diminished capacity to effectively learn about the minority class.

Finally, we would point out that stacking conventionally involves using only the predictions from different models as input to a final model, usually a simple linear model (e.g., Sill, 2009; Gunes et al., 2017). Including inputs may introduce redundancy that can negatively affect model performance; however, we have tested this approach. An additional analysis in the supplemental material now shows that when the stacked model is additionally exposed to the inputs the lower-level models were trained on, it performs worse (page 30,32).

7. Finally, I would be a little more cautious than the authors with the claim of near-term practical exploitation. This work depends on time-consuming pre-processing which is semi-automated at best, and which might well be impossible without high resolution, research quality scans. The authors might be confident that stroke clinics can be convinced to take such scans as standard, but I would find that confidence surprising. And if the scans won't be available, we will be stuck with much lower-quality and variable scans, which could well frustrate the kind of analysis used here.

Author response: *This is a fair criticism and we have made our claims in this regard more subtle (page 22).*

However, even taking all those gripes into account, I think this is a good paper - and particularly good given the promise of open-access data (which I will definitely aim to take up). Can the authors confirm whether they will also share analysis code? There are a lot of steps to this analysis that many of the rest of us would do well to follow, and could follow faster with their help!

Best wishes,

Thomas Hope

Author response: *We appreciate this and would gladly help however we can to make sure any interested researcher can run our pipeline from preprocessing to model training and testing.*

Our Aphasia Recovery Cohort (ARC) database is publicly available and contains raw data. We have also shared the data as preprocessed for this manuscript using a DOI that is currently

private. We planned to make this public upon manuscript publication but are happy to provide the private link in the meantime: <https://figshare.com/s/46012728175af8029c7b>

We have also updated our code availability section (page 26) to provide more comprehensive references for the code that we have used, beyond just the deep learning models. We hope our new edits provide sufficient resources and we would be glad to continue working with any researcher that needs help setting up the analysis pipeline or adapting it for their own work.

Reviewer #2 (Remarks to the Author):

This study compared two methods for aphasia prediction (defined binary as severe and non-severe). CNN on whole brain morphometry (based on segmented tissues) and lesion where compared with SVM (both linear and non-linear). The study population was 231 patients with chronic stroke. CNN achieved higher accuracy and F1 scores than SVM (but not necessarily other model performance metrics). Linear SVM performed better than non-linear, and additional dimensionality reduction didn't improve its performance. Importantly when linear SVM was trained on the latent features learned by the CNN, it performed identical to CNN. Using saliency map, authors suggest CNN leveraged widely distributed patterns of brain atrophy to predictive aphasia severity, whereas the SVM focused on the area around the lesion. However, there were many inter-individual variations. Identification of individuals with severe aphasia relied on features contralateral to the lesion, whereas identification of nonsevere aphasia relied on ipsilateral features.

The method is very well described and appears robust (although this reviewer is not an expert in CNN specifically). The findings are in keeping with the intuition and emerging evidence that whole brain health (in this case segmental atrophy), is associated with post stroke recovery, and extend beyond language specific regions. Using CNN for aphasia prediction is novel and I would like to see it published. Such methods, together with larger studies, are important for developing better aphasia prediction models that help with personalised delivery of care. However, some of the implications may have been over-stated, and there are some errors/clarifications that need to be addressed.

Author response: *We are grateful to the reviewer for the encouraging remarks and the constructive feedback. We believe we have effectively addressed the points raised, but remain open to additional modifications if needed.*

Introduction:

The motivation for focusing on SVM was not clear. Elsewhere in the methods it was stated that "in our experiments, logistic regression and decision trees performed just as well."

Author response: *Apologies, we can see how this sentence is confusing. The reference to logistic regression and decision trees refers to the process of building a simple model used to stack the predictions of the SVM and CNN models together. Here, we are just clarifying that the particular choice of algorithm for stacking predictions made no difference. This belongs in the supplemental material, and we have moved it there (page 30) and explained what an 'experiment' was in this context (i.e., qualitative inspection of performance measures across one repeat of cross-validation).*

We have clarified that we determined to use SVMs before any analysis took place because they are indisputably the most prevalent classical machine learning model in both neuroimaging and stroke neuroimaging (as shown in the PubMed search in Figure 2, now Figure S9). We have added clarification of this motivation to the introduction on page 4 and it is additionally mentioned in the methods on page 8 (lines 345-346).

When discussing contributions of domain general brain regions to recovery the following key work should be discussed:

- Dynamics of language reorganization after left temporo-parietal and frontal stroke. Brain 2020.
- Domain-general subregions of the medial prefrontal cortex contribute to recovery of language after stroke. Brain 2017.
- Task-induced brain activity in aphasic stroke patients: what is driving recovery? Brain 2014.

Author response: *These are excellent references, thank you for pointing us to them—we have happily added them to round out our discussion of brain regions associated with recovery on pages 2 and 24-25 (intro and discussion), and have tried to emphasize that some of the morphometry patterns we have decoded may reflect the contribution of these domain-general regions to aphasia severity.*

Figures:

There are two figures labelled 3. Please correct the labelling and double check all in-text references to all the figures.

Author response: *Thank you for catching this, this has been fixed.*

Figure 2 is better placed in supplementary material near the Sankey diagram, as its not fundamental to the study.

Author response: *Agreed—we have moved this figure to page 40.*

Wrong figure is cited: A histogram of WAB-AQ scores across participants is presented in Figure 1B.

Author response: *Fixed reference to histogram on page 5.*

For Figure 6A, comparing saliency map (Grad-CAM++). Can the authors confirm that the lesion overlap map is not obscuring the view of underlying voxels that have feature importance for this method. Its hard to appreciate any overlay effects. (ie no green/dark blue covering up yellow).

Author response: *We can confirm that there is no overlap in effects that is being obscured in this figure (i.e., no peaks in saliency in the left hemisphere). We would emphasize Figure 8, where it is more clearly appreciable that Grad-CAM++ is on average significantly higher in the right hemisphere than the left for patients with severe aphasia. The visualizations in Figure 6 are qualitative, showing general trends in peak saliency. We have uploaded these brain maps to a public NeuroVault repository as well: <https://neurovault.org/collections/16012>*

Figure 7: Its helpful to add in the CNN violin plot for comparison.

Author response: *We would be happy to add this if necessary. However, we would prefer not to duplicate information presented in other figures to avoid lengthening/complicating the manuscript. Further, we feel adding CNN performance to this figure would take the focus off the main finding we would like to highlight.*

We see the main purpose of this figure as emphasizing that the saliency maps that can capture the special spatial properties learned by the CNN lead to better predictions than those that don't. The text for this analysis (page 17) links to supplemental material that shows violin plots comparing CNN performance to SVM models trained on the CNN features. The supplemental material also contains direct statistical comparisons between these models (pages 30-31). Compared to the saliency maps, these lower-dimensional features more directly capture the representations learned by the CNN and therefore are a better alternative to Grad-CAM++ for understanding and testing whether an SVM can more successfully exploit information learned by the CNN.

Figure 8: Left Right orientation of brain slices are opposite to the rest of the figures. Can this be reversed for consistency?

Author response: *Thank you for catching this, we have reversed the figure.*

Figure 9: Panels A,B, C are not labelled .

Author response: *The compiled manuscript on our end is not missing the labels but we will make sure to upload higher resolution figures separately during submission so that the panels can be read more clearly in the case that they are being cut off somewhere during the compilation process.*

Results:

Even though the authors state they are mainly interested in the F1 score, the 'precision' of the model in predicting severe vs non severe aphasia was barely above chance 0.59. This doesn't seem very effective.

Author response: *We appreciate this concern and agree that precision is a useful metric for evaluating our model. This is why despite justifying an overall emphasis on model recall in the text (page 7, lines 293-295 and see new edits on this page), we chose to focus on a balanced measure between recall and precision.*

We argue that a model that aims to predict individuals who have or will go on to have severe aphasia should minimize false negatives. As such, focusing on the precision of the model alone provides an incomplete picture. This is doubly the case in our application as there is a class imbalance, and the more frequent class is the negative one (non-severe aphasia).

When there is class imbalance, focusing on precision can be pessimistically misleading. Models tend to predict the majority class more frequently in a general strategy to minimize loss. Thus, even if a model is relatively good at identifying the minority class (severe aphasia), the inflated

number of false positives due to a bias stemming from class imbalance can reduce the precision score. Achieving an average precision of 0.59, despite the imbalance, indicates that the model is effectively identifying a substantial proportion of severe aphasia cases.

We don't disagree that the precision score highlights much room for improvement (and have now pointed this out more clearly with discussion in the limitations section on page 25), only that the score we attained reflects an overall poorly performing model. To that end, we further point out that our thorough permutation analysis shows the model performs significantly better than chance. It is rare to see this kind of model confirmation in deep learning due to the computational costs, but we have chosen to perform this analysis to provide very clear support of the models' success based on the standard benchmark in machine learning studies like ours (i.e., using random chance as the baseline). Additionally, we would highlight that the F1 score gives preference to models that have more similar precision and recall, and the attained F1 score has been described to be in the "quite good" range in other applications with similar precision and recall (e.g., in a well cited review of classification metrics by Grandini and colleagues, 2020).

Can the authors comment on why SVM outperform CNN in predicting non-severe aphasia accuracy? Evident in Figures 4 and 5. Relatedly when discussing the Stacking approach, the authors limit the discussion to F1 scores, but do not consider significantly improved performance of CNN in other metrics when stacked with SVM.

Author response: *Certainly, we have added clarifications on pages 13 and 14.*

The CNN model is making more false positive errors for the minority class as indicated by the marginally lower precision score (see response to previous comment as well). However, F1 and balanced accuracy show the CNN performs slightly better, reflecting better performance when both classes are given equal weight. Consistent with this, SVM performs slightly better than CNN on majority class accuracy while CNN performs a lot better on minority class accuracy. In other words, CNN achieves better overall performance because it improves on the harder to predict severe class accuracy more than SVM improves on non-severe accuracy. It's important to point out that both models had loss penalized by inverse class frequencies so the fact that the SVM did not exploit the same strategy as the CNN suggests it was not able to learn as much about the minority class.

We have also expounded on our explanation of the model fusion results, explaining how stacking does improve predictions marginally (but insignificantly) by attempting to boost majority class accuracy relative to the CNN (page 14). Note, we improved our procedure for staking, see additional details on page 9 of the manuscript and response to reviewer 1 on pages 4-5 of this document.

Is it valid to compare the saliency maps for the non-severe patients to the meta-analysis data of neuroimaging literature when we see the CNN has a low non-severe aphasia accuracy?

Author response: *The saliency and decoding analyses serve to explain and contextualize the patterns learned by the model so we believe this analysis is justified in the way we present it.*

We acknowledge the concern that the distribution of feature importance may change as a model becomes more accurate, and therefore explanations of its performance will be different. We

have updated the discussion to reflect this concern (page 25). This is a challenging issue that broadly affects application of machine learning to biological problems.

Here, however, we would point out that our core conclusions about the saliency and decoding results do not hinge on individual features. The decoding analysis considers the whole brain pattern of feature importance. While this pattern might shift with model accuracy, our main finding is that a substantial proportion of altered morphometry patterns are strongly associated with patterns of activity reported outside of studies related to language. This is consistent with other sources of evidence that support that brain integrity outside the stroke lesion can contribute to aphasia severity.

We would also point out that as our approach was novel, there is no alternative model capturing spatial dependencies that we can compare to that may either be more accurate or provide an alternative explanation (i.e., feature importance assignment) for prediction of aphasia severity from morphometry data.

In line with our previous comments, we would also stress the importance of evaluating the model holistically instead of focusing on performance in a single class. For example, severe prediction accuracy for CNN is high but the model makes more predictions of severe aphasia in general. Thus, the models' learned representations for non-severe aphasia may technically be more 'accurate', but class accuracy might reflect the models' learned bias towards making a particular prediction.

“CNN models often correctly classified participants with medium-sized lesions that belonged to both severity classes”, Based on the figure looks like many medium lesions were misclassified. This is misleading.

Author response: *Apologies, we have removed this sentence to avoid any confusion. Our overall point was that medium sized lesions are observed in both aphasia classes, and correspondingly, the CNN model does not always associate this lesion size with one particular class (i.e., providing some suggestion that the CNN is not simply using lesion information)*

Discussion:

The prediction of the CNN model is very binary: severe vs non severe aphasia. On a philosophical level, what do the authors see as the actual benefit of a model that can predict severe aphasia from brain imaging when one can do that prediction by simply spending a couple of minutes talking to the patient? What is the impact of such tools? I don't see them impacting treatment decision either, as treatments are impairment based, rather than based on broad diagnostic categorisation.

Therefore I don't think the results shown support this statement : “it may soon be possible to deploy similar models at point-of-care, where prediction of aphasia severity may help healthcare professionals prepare patients for their anticipated outcomes and inform interventions.”

Author response: *This is a fair point and we've revised our statements to be more subtle, including in the referenced quote on page 22 (and additional edits on page 23).*

We don't expect our particular model to have direct clinical utility. What our model shows is that patterns of brain integrity outside of the lesion and outside of the putative language network contribute to aphasia severity. This suggests that it may be possible to train similar models (i.e., CNNs because accounting for spatial dependencies was crucial to identifying such patterns in our data) with more direct clinical utility--for example, by predicting whether someone is likely to go on to have severe aphasia based on whole brain integrity outside of the stroke lesion on hospital intake scans. These are the types of models that may help healthcare professionals and could possibly provide insights unavailable when prognosticating from behavioral data alone.

Minor:

- Introduction: "interhemispheric territory" > best described as midline brain structures
- The term Neural network is confusing eg : "subtle patterns of atrophy across the neural network". Whole-brain level atrophy would be more clear. Network often is used in context of structural and functional connectivity
- Reference 87 is repeated

Author response: *Thank you for catching these, we have made the suggested revisions.*

Reviewer #3 (Remarks to the Author):

1. Brief summary of the manuscript

This study investigates the potential of deep learning models to predict aphasia severity at the chronic stage after stroke from brain morphometry. The authors show that Convolutional neural networks (CNN) outperform classical machine learning methods such as Support Vector Machine (SVM) in predicting aphasia severity. Information from the brain morphology, especially beyond the lesion location, seems to be a significant contributor to the CNN model performance.

2. Overall impression of the work

This manuscript is a very comprehensive and high-quality investigation. To my knowledge, no similar work of this scale has been published in the field of post-stroke aphasia prediction and I commend the authors for the rigor and extensiveness of the analyses performed and the clarity of the description of these analyses. This is an important piece of work that provides a compelling precedent for future ML studies on this topic. I have a few suggestions detailed below to enhance the interpretability of this work and facilitate translational applications and I'll mention the four most important here.

Author response: *We thank the reviewer for this encouraging feedback as well as the very helpful comments below. We hope we have been able to address them but are open to further changes.*

One important limitation, which should be highlighted but does not reduce the quality of this paper, is that this work is cross-sectional and therefore does not predict post-stroke aphasia outcomes but aphasia severity at the chronic stage using data obtained at the same stage. This is a common caveat of papers on this topic and the word "predicts"/"prediction" may mislead some readers to think that it takes information at one point in time and predicts information at a future point in time. Although there are not many alternative verbs in ML to describe the results

of these models, this limitation should be highlighted in the discussion as it may have significant implications on the capacity of this type of work to be translational.

Author response: *We thank the reviewer for drawing our attention to this potential source of confusion and have tried to make clear that this work does not predict aphasia severity from earlier timepoints. We've made changes to this effect in the abstract (line 19) and introduction (lines 150 and 152). We have also attempted clarification in multiple changes of the discussion on pages 22 and 23. We hope these are sufficient but are open to further changes and do not wish to mislead readers.*

As I mentioned, this work is already a great example to follow for future studies that would explore longitudinal data but this difference needs to be clear because we might expect different findings and interpretations due to changes in brain morphometry (e.g., including the size of the lesion) over time across post-stroke recovery stages if the input of the models are data acquired at the acute stage predicting outcomes at the chronic stage (which is the ultimate goal of this type of work, as stated in “prediction of aphasia severity may help healthcare professionals prepare patients for their anticipated outcomes and inform interventions”).

Author response: *We appreciate this point and to avoid any confusion have also clarified the referenced sentence as part of the edits (pages 22-23). We have also added discussion of the nuanced point that features may be different in models like ours that are trained on longitudinal data (page 25).*

Another point is that the authors highlight that aphasia severity is better predicted by CNN because of the importance of the information beyond the lesion site. If that was the main hypothesis (information beyond the lesion site may improve prediction of aphasia severity), I wonder why the authors did not compare two types of input to the CNN models: one model with only the lesion site information (location and size) and one with lesion information + information from the rest of the brain. While comparing CNN and SVM may lead to this type of comparison indirectly (CNN architecture being more sensitive to whole-brain morphology patterns compared to SVM), this interpretation is indirect and mostly based on post-hoc observations when interpreting the saliency maps. Adding this analysis to the paper would, in my opinion, improve the interpretations of the authors.

Author response: *We concur that evaluating feature importance is an indirect analysis that nevertheless clearly demonstrates that the CNN leverages whole brain morphometry patterns. We did not directly test for the influence of features outside the lesion in the main text because interrogating model saliency was already necessary for us to determine whether the CNN exploited spatial dependencies—our primary analysis of interest.*

Nonetheless, the reviewer raises a very important point and we would like to highlight that our supplemental ablation experiments more directly show that features outside the lesion matter and these results are referenced in the main text. In these analyses (Figure S5), we trained a SVM on high dimensional CNN saliency maps, removing either the left lesioned hemisphere or the right intact hemisphere. Consistent with the saliency analysis we present in the main text, we found that using both hemispheres resulted in significantly better performance and using either hemisphere alone produced comparable model scores (i.e., SHAP shows the CNN attends to the left hemisphere while Grad-CAM additionally shows it tends to focus on right hemisphere spatially dependent patterns).

To further confirm these results, we reran the CNN model while removing all information other than the lesion. While this model generally outperformed SVM trained on all features, it performed slightly yet significantly worse than the CNN model from the main text, $p = 0.02$. A visualization of this direct test has been added to the supplemental material (see additions to Figure S5 and page 34). We now reference this test in the main text (page 17). This finding is consistent with our interpretation that spatially dependent morphometry patterns outside the lesion provide a small but meaningful boost to model performance by identifying unique patterns.

Another caveat is that the authors focus all their interpretations and hypotheses around the impact of brain atrophy on aphasia severity. However, as the authors suggest themselves in one paragraph of the discussion, the CNN could also capture other type of information related to 1) the stroke in regions outside of the lesion or 2) (not mentioned by the authors) overall brain health (e.g., lacunes, perivascular spaces). I may have missed something that would relate specifically the CNN models to atrophy but if there is not a clear direct relationship between the two, the authors may need to rephrase their introduction and discussion to not limit their main interpretations to patterns of atrophy and extend it to overall brain health (“morphology patterns outside the lesion” is also a good alternative used by the authors in some places).

Author response: *We had in some places clarified that the patterns captured by the CNN may include precisely these types of phenomena, however, we agree that this could be clearer and have changed text throughout to this effect (e.g., page 3). We have also made changes throughout the manuscript to avoid using the phrase “atrophy” entirely in the context of our results. We retain references to atrophy where we believe it is clear that this is an example of the morphometry patterns that may be captured (e.g., changes to the abstract or Figure 1).*

Related to the previous point, statements like these two seem speculative: “Our findings demonstrate that individuals with severe aphasia additionally have extensive atrophy of the right hemisphere” / “CNN identified atrophy in the right hemisphere as a strong predictor of severe aphasia” (the manuscript only shows that information from the anatomy, in a broad sense, not just atrophy, of the right hemisphere is important to classify individuals with severe aphasia).

Author response: *This is a fair point, we have made our statements more subtle here and throughout the manuscript so that we are not making specific claims about atrophy alone.*

Finally, although accuracy was not the main measure to assess model performance, the authors may add more nuance in their interpretations and could mention in the results and discussion the non-trivial higher performance of the stacked model for the non-severe group in terms of accuracy (Figure 5B) as it seems to indicate that SVM may actually capture some information about the non-severe group that CNN does not.

Author response: *We have added commentary on this in the results, explaining that stacking did provide a boost to majority class predictions from the lower-level SVM model that contributed to an overall marginally better performing model. It is worth pointing out, however, that the stacked model did not perform significantly better, suggesting that the SVM did not clearly identify substantially different predictive information (note, we improved our procedure for staking, see additional details on page 9 of the manuscript and response to reviewer 1 on pages*

4-5 of this document). For example, the SVM's tendency to more frequently predict the majority class may reflect a bias that serves to improve accuracy when the model is unable to learn as much about the minority class. That SVM did not identify substantially different information is corroborated by the overlap in SVM and CNN model saliencies when model attention to spatial dependencies was ignored (e.g., Figures 6 and 8). We have clarified in the discussion and throughout the results that the SVM might capture unique information, yet our multiple analyses, which aimed to find a clear effect and readily apparent effect, did not provide evidence to support this (for example, see pages 25-26).

3. Additional specific comments

- In the introduction, please add citations related to this sentence: "Beyond lesion size, the spatial location of stroke injury is predictive of aphasia severity".

Author response: *We have added a reference for this.*

- The paragraph following "Within this framework, chronic aphasia tends to be less severe when core language specific regions, ..." in the introduction does not explain accurately the framework cited. The recruitment of domain-general regions and its role in this theoretical hierarchy has not been described in this framework. This theory only mentions secondary centers in the ipsilateral network and perilesional areas. I agree however that, since this framework, multiple studies have hypothesized a role of domain-general regions in aphasia recovery alongside right-hemisphere regions. I suggest the authors provide more recent citations that support these hypotheses and rephrase this paragraph to match the right hypotheses with the respective papers and avoid claims on a hierarchy between the recruitment of domain-general and right-hemisphere regions (or provide citations for this).

Author response: *Thank you for catching this, we have edited this paragraph on page 2 to more clearly separate the referenced framework from more recent work implicating domain-general regions in recovery. However, we would clarify that the referenced framework does assign a role to contralateral homotopic regions and we hope the additional references help support this overall point (i.e., more right hemisphere involvement in more severe cases).*

- I may have missed this information but if it is not there, the authors need to mention the range and average of months post-stroke onset for the population included in these analyses.

Author response: *We missed this information and have added it to the revised manuscript (page 4).*

- The authors may want to briefly explain what filters and contrasts mean in the context of CNN for clarity.

Author response: *We can see how these terms would be confusing and have clarified them in the introduction (pages 3 and 7).*

- The numbering of the figures needs to be rearranged (there are two figures 3, and this

sentence does not point to the correct figure: “A histogram of WAB-AQ scores across participants is presented in Figure 1B”)

Author response: *Thank you—fixed.*

- The authors mention in the discussion that downsampling to 8mm voxel size is a limitation but do not explain why they made this choice in the first place as it seems other options were available. It would be good to add this explanation.

Author response: *We have clarified on pages 5 (methods) and 25 (limitations) that downsampling was necessary to make the cross-validation scheme with deep learning computationally tractable. While higher resolution data may produce a more accurate model, it necessitates a greater degree of CNN model complexity (i.e., more computations per layer as the feature maps become larger), making model tuning and training more difficult to perform successfully.*

- In Figure 3 (the first one), the authors may want to simplify and clarify the box related to VGG-style CNN by reducing the amount of text and highlighting the important information.

Author response: *We appreciate this but are not sure there is a better way to present the network architecture visually. Our visualization approach, including the text, is in line with other studies (for example, see Karakis and colleagues or reference 22 from the main text).*

The challenge here is that while the networks simply increase in complexity from left to right, both the composition of each colored block and the properties of individual layers can differ. While we summarize the logic of how they differ in the methods, it's difficult to visualize these rules so we feel it is helpful to have this representation.

We are reticent to fully explain the network architecture in the figure caption as it may take up substantial space, but have made reference to the methods section that provides further details and have inserted some explanation in line with the above description for more clarity (page 6).

Citations

Grandini, M., Bagli, E., & Visani, G. (2020). Metrics for multi-class classification: an overview. *arXiv preprint arXiv:2008.05756*.

Güneş, F., Wolfinger, R., & Tan, P. Y. (2017, April). Stacked ensemble models for improved prediction accuracy. In *Proc. Static Anal. Symp* (pp. 1-19).

Sill, J., Takács, G., Mackey, L., & Lin, D. (2009). Feature-weighted linear stacking. *arXiv preprint arXiv:0911.0460*.

Reviewer #1 (Remarks to the Author):

The authors have answered my queries satisfactorily. In my view, they have also provided good answers to the queries advanced by the other reviewers. So I am happy to approve the manuscript.

Reviewer #2 (Remarks to the Author):

The authors have comprehensively addressed all the points I had raised. I look forward to seeing this important work published.

Fatemeh Geranmayeh

Reviewer #3 (Remarks to the Author):

Thank you for answering my questions and addressing my concerns. I look forward to seeing the published version. This is a great manuscript.